# Impact of Interfaces, and Nanostructure on the Performance of Conjugated Polymer Photocatalysts for Hydrogen Production from Water

**DOI:** 10.3390/nano12234299

**Published:** 2022-12-03

**Authors:** Ewan McQueen, Yang Bai, Reiner Sebastian Sprick

**Affiliations:** 1Department of Pure and Applied Chemistry, University of Strathclyde, Thomas Graham Building, 295 Cathedral Street, Glasgow G1 1XL, UK; 2Institute of Materials Research and Engineering, Agency for Science Technology and Research, #08-03, 2 Fusionopolis Way, Innovis, Singapore 138634, Singapore

**Keywords:** photocatalysis, solar fuels, hydrogen generation, water splitting, carbon dioxide reduction, conjugated polymers, covalent organic frameworks, conjugated microporous polymers

## Abstract

The direct conversion of sunlight into hydrogen through water splitting, and by converting carbon dioxide into useful chemical building blocks and fuels, has been an active area of research since early reports in the 1970s. Most of the semiconductors that drive these photocatalytic processes have been inorganic semiconductors, but since the first report of carbon nitride organic semiconductors have also been considered. Conjugated materials have been relatively extensively studied as photocatalysts for solar fuels generation over the last 5 years due to the synthetic control over composition and properties. The understanding of materials’ properties, its impact on performance and underlying factors is still in its infancy. Here, we focus on the impact of interfaces, and nanostructure on fundamental processes which significantly contribute to performance in these organic photocatalysts. In particular, we focus on presenting explicit examples in understanding the interface of polymer photocatalysts with water and how it affects performance. Wetting has been shown to be a clear factor and we present strategies for increased wettability in conjugated polymer photocatalysts through modifications of the material. Furthermore, the limited exciton diffusion length in organic polymers has also been identified to affect the performance of these materials. Addressing this, we also discuss how increased internal and external surface areas increase the activity of organic polymer photocatalysts for hydrogen production from water.

## 1. Introduction

The development of photocatalysts that drive the generation of solar fuels has been an area of interest for many years. Conjugated polymers as photocatalysts for hydrogen production from water were first reported by Yanagida’s group in 1985 (Figure 1) [1]. They were able to show that poly(*para*-phenylene) was able to facilitate proton reduction in the presence of a sacrificial hole scavenger. Unfortunately, these early efforts were not significantly followed up for more than two decades until the first report by Wang et al. on carbon nitride as a photocatalyst for sacrificial hydrogen and oxygen production [2,3]. Following this and the publication of photocatalytically active conjugated polymer networks [4,5], the field has significantly grown with many research groups working towards conjugated materials that facilitate overall water splitting with high efficiency.

While inorganic photocatalysts have been studied far more extensively than their organic polymeric analogues [6], organic materials are an attractive alternative as their structure can be easily tuned using synthetic approaches. This also allows the systematic tuning of the materials’ properties, such as band gap and band positions, polarity, particle size and porosity to name but a few. The tuning of the properties provides a large scope for performance optimization, while the potential of conjugated polymers to be solution-processible is highly beneficial for industrial manufacturing towards application at scale [7,8]. Fundamental knowledge in organic photovoltaics (OPVs) can be built on in photocatalysis, as some of the underlying fundamental processes are similar. However, the review will discuss a number of processes which have different requirements in water splitting or are even irrelevant in OPVs.

A multitude of different methods have been used to synthesize polymer photocatalysts (Figure 2), with the most widely used method being the metal-catalyzed Suzuki−Miyaura coupling reaction. This reaction provides high yields and can be used for the coupling of a wide range of multifunctionalized halogen monomers with boronic acid/acid ester monomers [9,10,11]. Suzuki-Miyaura coupling, and closely related coupling methods such as Sonogashira coupling [12,13,14] and Stille-coupling [15,16], result in residual palladium being left in the material. This residual palladium is difficult to remove and has been shown to act as a catalyst for proton reduction [17,18]. Nevertheless, palladium can act as a recombination center for the backward reaction of water splitting. This could potentially provide severe drawbacks in the future in overall water splitting systems as it limits their activity [19], and thus requires additional modification in the future [20]. Methods that do not require metal-catalysts for the coupling reactions to form polymeric photocatalysts have also been extensively studied: A good example being the ionothermal method for the condensation of suitable precursors to synthesize CTFs [21,22] and C_3_N_4_ [2].

Many of the materials obtained from metal-catalyzed and metal-free approaches show poor long-range order, which is often thought to be detrimental to charge transfer and separation in photocatalysis. As such, materials with much higher degrees of crystallinity, in particular COFs, have been developed. These are obtained through reversible condensation reactions of suitable monomers under thermodynamic control [26]. Following on from the first reports of this material class, azine COFs obtained through condensation reactions were used as photocatalysts for sacrificial hydrogen production from water [27]. Photocatalytically active COFs with even higher degrees of crystallinity are obtainable through pre-organization of the building blocks using a reversible and removable covalent tether, followed by confined polymerization [28]. Furthermore, carbon nitrides with high degrees of crystallinity have also been obtained, using ionothermal synthesis with metal salts in addition to the reaction solvent. These materials have shown high levels of activity for photocatalytic overall water splitting by controlling reactive facets [3].

A greater understanding has been gained in recent years on how factors such as light absorption [15], the driving-forces of the half-reactions [9,29], exciton separation [30,31,32,33] and polaron formation [34], conjugation length [35], crystallinity [27,28,36], co-catalysts [17,19,37,38], surface wetting, particle size [36], and surface area [37] affect the photocatalytic performance of conjugated polymer photocatalysts. Reviews on conjugated polymer/polymer network photocatalysts [8,23,24,25,38], as well as reviews on covalent organic framework photocatalysts [39,40,41,42], provide an overview and discuss some of these factors [43,44]. This review specifically focuses on the interface with water, and the nanostructure of the interface, followed by briefly discussing the current understanding of the interface between the metal co-catalyst and polymer photocatalyst. The aim is to draw conclusions that are applicable to all polymer material classes, providing insight that will be useful to all researchers studying polymeric materials for application in photocatalytic hydrogen production.

## 2. Mechanism of Photocatalytic Water Splitting

In photocatalytic water splitting reactions suitable semiconductors are placed in dispersions with water and irradiated with light producing hydrogen and oxygen simultaneously in a 2:1 ratio. Light with sufficient energy promotes electrons from the highest occupied molecular orbital (HOMO) to the lowest unoccupied molecular orbital (LUMO) of the photocatalyst resulting in the formation of electron-hole pairs, so-called excitons (Figure 3). The band gap energy has to be larger than 1.23 eV to fulfil thermodynamic requirements, however, in most cases it should be larger than 2.0 eV considering kinetic barriers. The splitting of the exciton allows charges to flow to active sites that then facilitate the reduction of protons and the oxidation of water liberating both hydrogen and oxygen. Exciton separation into individual charges is challenging in organic semiconductors as the electrostatic binding energy of the electron–hole pair, originating from the Coulomb interaction between the photoexcited electron and hole, is greater than the thermal energy available at room temperature (approx. 26 meV). Overcoming the exciton binding energy requires additional energy, thus lowering the performance capacity of polymer photocatalysts. To overcome this, strategies such as employing donor-acceptor heterojunctions have been explored and has led to a large enhancement in photocatalytic activity in the literature [30,33].

Fundamentally, the HOMO and LUMO levels have to be sufficiently positive and negative for the water oxidation and proton reduction reactions, hence the bandgap must straddle the water oxidation and proton reduction potentials respectively. Nevertheless, to facilitate the charge transfer of electrons and holes from the semiconductor to protons and/or water, metal catalysts are usually required [45,46].

A challenge for organic photocatalysts is the water oxidation process. Due to its sluggish kinetics involving a four-electron transfer process, water oxidation remains very challenging. On top of this, attaining organic semiconductors with sufficiently low-lying HOMO levels for water oxidation remains challenging [6,47]. Therefore, in many cases sacrificial hole scavengers are oxidized instead of water [27,48]. This allows these systems to be better understood by decoupling the challenges of the hydrogen evolution reaction from the water oxidation reaction. Yet beyond the consideration of the redox potential of the sacrificial reagents relative to the photocatalyst HOMO level, no explicit guidance exists [49]. Furthermore, it is important to note that scavengers ultimately have to be replaced with water to make the process energetically viable in the future.

## 3. Effect of Interfaces in Polymer Photocatalysis with Conjugated Polymers

For materials to perform water splitting, or even photocatalytic half-reactions, the interaction with water is key on the microscopic level, but also when considering the dispersion of the macroscopic particles during photocatalysis. This is of course very different when comparing to OPVs where no liquid interface is present. Wettability is associated with the degree of contact involving a liquid with a solid surface through intermolecular interactions, which can be influenced by polarity and surface morphology [7]. On the molecular level water and protons have to reach active sites on the photocatalyst with sufficiently high rates to allow the extraction of charge carriers and production of hydrogen and oxygen. Conjugated polymers by their very nature are apolar, especially with respect to examples of inorganic photocatalytic semiconductors such as metal oxides and chalcogenides [50], thus water is less likely to interact well with the surface of these materials [51]. As a result of the lack of interaction at the interface, polymer particles are likely to poorly disperse and rise to the surface due to their low density (Figure 4a,b).

Considering the interface, it is not unsurprising that poly(*para*-phenylene) P1 as the first example of a polymer photocatalyst for hydrogen production from water [1], showed only modest activity under UV light using diethylamine as a hole scavenger. A later study showed that using diethylamine as the organic hole scavenger increased the dispersibility of the photocatalyst, and that using Na_2_S as a non-organic hole scavenger resulted in the material becoming inactive due to limited wetting and poor dispersion in water [48].

Studies that followed also did not necessarily consider wettability when studying organic polymers as photocatalysts: For example, the report of copolymerized CMPs of pyrene and benzene monomeric units in 2015 (CP-CMP1–CP-CMP15, Figure 5) [5]. The materials had measurable, but low activity for photocatalytic hydrogen production from water in the presence of diethylamine compared to more established photocatalysts, such as carbon nitrides at the time. These materials are also apolar due to the nature of the pyrene building blocks. This results in poor wettability and a limited interface despite the high levels of nitrogen accessible surface areas in these materials (up to 1700 m^2^ g^−1^). Other studies at this time, such as the report of phenylene CMPs [52], benzothiadiazole CMPs [53], and hypercrosslinked spiro-fluorene [54] also did not consider the wettability of these materials and thus have only modest sacrificial hydrogen evolution rates.

A significant breakthrough was found when heteroatom-containing materials were studied. Linear conjugated copolymers with fluorene-type linker units and varied atoms at the bridgehead showed significantly increased photocatalytic activity for sacrificial hydrogen generation from water, compared to poly(*p*-phenylene) (Figure 6, P1) [56]. The introduction of a fluorene based monomer results in a more planarized structure which has been suggested to reduce the propensity for electron-hole recombination, as well as increasing charge carrier mobility [57,58]. Interestingly, on top of this, the alteration of the bridgehead atom allows polymers of varying polarity in the backbone to be studied.

Particularly, the introduction of dibenzo[*b,d*]thiophene and dibenzo[*b,d*]thiophene sulfone units in P6 and P7 resulted in significantly increased activity. The P6 activity under visible light irradiation (λ > 420 nm, 300 W Xe light source, triethylamine scavenger) is nearly 20 times greater than that of P1, and its activity under broad-spectrum irradiation (λ > 295 nm, 300 W Xe light source, triethylamine scavenger) is comparable to that of TiO_2_ (rutile) loaded with 1 wt. % Pt. P7 produced an even more substantial rise in activity with an AQY of 7.2% compared to 0.4%. The difference in optical gap and thermodynamic driving force for proton reduction when considering P6 and P7 is not significant and therefore it was suggested that the increased activity may be attributed to differences in either charge carrier and mobility, or particular surface chemistry [56]. A factor that was not considered at the time was wettability.

Describing wettability in isolation, it is likely that the more polar backbone in P6 and P7 results in higher wettability of the polymer surface with respect to the aqueous environment [7]. These factors described above likely coalesce to enhance catalysis, which highlights the difficulty of studying one property in isolation via synthetic/structural modification for many photochemical reactions. A major difference for this series of linear conjugated polymers with respect to CMPs discussed previously is the lack of microporosity, thus any wettability assessments of this linear series is likely to do with direct surface intermolecular interaction of the atoms that are present on the polymer backbone with the water/sacrificial reagent mixture [5,56].

## 4. Strategies for Increased Wettability in Linear Conjugated Polymer Photocatalysts

The following sections will discuss the incorporation of polar units into the materials’ aromatic core as well as into the side-chains to increase the polarity and thus wettability. Increased external surface areas, as well as internal surface areas, will also be discussed as this in principle increases the availability of active sites which is often linked to an increase in photocatalytic activity.

### 4.1. Polar Core Units for Increased Wettability

Following on from early studies it became clear that wettability of the photocatalysts contributes significantly to their activity: Conjugated microporous polymers of perylene bisimide, spirofluorene, perylene, and di(thiophen-2-yl)benzo [1,2-*b*:4,5-*b*′]dithiophene with biphenyl as a comonomer were all outperformed by bipyridyl containing analogues which showed significant increase in their photocatalytic rates, e.g., the rate of the spirofluorene polymer with biphenyl increased from 1.39 µmol h^−1^ to 5.11 µmol h^−1^ for the bipyridyl material (λ > 420 nm, 150 W Xe light source, triethylamine scavenger) [59]. Similarly, di(thiophen-2-yl)benzo [1,2-*b*:4,5-*b*′]dithiophene polymers with nitrogen-containing heterocycles showed increased sacrificial hydrogen production rates compared to non-polar aromatic analogues [60], which was also found to be the case for linear poly(*para*)phenylene analogues [61]. It is important to note that differences in other factors, such as light absorption, thermodynamic driving forces, and excited state lifetime also play an important role making it difficult to isolate wettability as a single factor.

Following on from the first report of dibenzo[*b*,*d*]thiophene sulfone polymer photocatalysts, a study directly compared poly(*p-*phenylene) (P1), the co-polymer of phenylene and dibenzo[*b*,*d*]thiophene sulfone (P7), and the homopolymer of dibenzo[*b,d*]thiophene sulfone (P10) to identify structure-activity relationships for sacrificial hydrogen evolution from water (Figure 7) [48]. Contact angle measurements were used to measure the wettability and found to be 88°, 67°, and 59° and for P1, P7, and P10, respectively. This suggests that the wettability of these polymeric materials increases with increasing dibenzo[*b*,*d*]thiophene sulfone content, which corresponds with an increase in polarity of the backbone. A high external quantum efficiency (EQE) of 11.6% at 420 nm was reported for **P10**, which is significantly higher than the EQEs of P1 (0.4%) and P7 (7.2%) at the same wavelength [48].

Molecular dynamics (MD) was used to predict the interaction of the surface with the aqueous environment (H_2_O/TEA mixture, Figure 8). Oligomer analogues used in the simulation showed that the SO_2_ groups tend to direct towards water molecules until a shell of water molecules surrounds it even in the presence of a sacrificial reagent domain, which is perhaps expected when considering the polarity of the sulfone group. The amphiphilic nature of P7 and P10 is highlighted since the less polar components tend to reside in the TEA domain, which is also the case for the P1 analog. This may suggest that this is a crucial difference which leads to the increase in photocatalytic efficiency with respect to P1. This polar environment is also suggested to facilitate acceleration of charge and proton transfer steps [48]. It can be deduced, at least for this series, that the influence of the polar sulfone group which leads to greater wettability of the polymer particles, is likely to work in harmony with the enhanced photo-induced intermediates and processes, which lead to greater photocatalytic activity for sacrificial hydrogen evolution from water. However, as is the case for many photochemical reactions, it is difficult to determine the exact extent of the contribution this phenomenon has towards enhanced catalysis. This study suggests that differences in thermodynamic driving forces, differences in charge carrier lifetimes of excitonic states (as determined by transient absorption spectroscopy; TAS) and even differences in solvation properties may all contribute to the observed differences in the photocatalytic data for P1, P7, and P10, respectively [48].

A later study showed that the incorporation of dibenzo[*b*,*d*]thiophene sulfone into pyrene networks increased the activity by nearly a factor of three compared to the phenylene analog [66]. A series of solution-processible fluorene co-polymers with increasing dibenzo[*b*,*d*]thiophene sulfone content have shown a similar trend, with polyfluorene being nearly inactive and the activity for sacrificial hydrogen production under visible light increasing from FS1 (EQE at 420 nm = 0.04%, 3% dibenzo[*b*,*d*]thiophene sulfone content) to FS5 (EQE at 420 nm = 2.07%, 50% dibenzo[*b*,*d*]thiophene sulfone content). Furthermore, it was found that the sulfone group does not only influence the interaction with water but also assists the formation of polarons and the charge transfer to palladium where proton reduction occurs [67]. Other studies have also shown that a complex interplay of different factors is important. For example, high throughput computational studies attempted to deconvolute properties associated with a library of organic polymer photocatalysts in isolation for sacrificial hydrogen generation, however no strong correlation between any isolated physical property and performance across the extensive library was determined [10]. This highlights the diversity in structure-activity relationships for organic polymeric photocatalyst materials. In a later study, it was shown that there is a relationship between the polarity extent of the polymer backbone and a contribution to enhanced photocatalytic activity [64]. In a series of copolymers involving heteroatom-substituted fluorene monomers, it was determined that the more polar moieties incorporated, the greater the interaction with water, hence an increase in wettability and thus hydrogen evolution activity. The polar environment aids stabilization of the charge-separated state after photoexcitation and hole scavenging which significantly enhances hydrogen generation [64].

The increased wettability and effects that are a result of the polarity of dibenzo[*b*,*d*]thiophene sulfone has resulted in a large number of follow up studies [64,68,69,70,71,72,73,74,75,76,77,78,79]. Some of these materials have shown very high sacrificial hydrogen evolution activity, with a dibenzo[*b*,*d*]thiophene sulfone-dibenzo[*b*,*d*]thiophene co-polymer reaching an EQE of 20.7% at 420 nm [10], and a co-polymer of dibenzo[*b*,*d*]thiophene with ethylene dioxythiophene even showing good sacrificial activity further into the visible light region (EQE = 13.6% at 550 nm and 7.3% at 600 nm) [80]. A co-polymer of dibenzo[*b*,*d*]thiophene sulfone with terthiophene achieved EQEs of 25.3% and 7.9% at 550 nm and 600 nm respectively, in water/*N*,*N*-dimethylformamide mixtures with ascorbic acid acting as the hole-scavenger [81].

Other fluorene-like polar units have also shown significant potentials: The closely related analog to dibenzo[*b*,*d*]thiophene sulfone, dithieno [3,2-*b*:2′,3′-*d*]thiophene sulfone (Figure 9), was used in conjugated microporous polymers with a pyrene comonomer showing quantum yields of 2.1% at 600 nm, and 1.1% at 650 nm, in water/*N*,*N*-dimethylformamide mixtures with ascorbic acid acting as the hole-scavenger [82]. Phenylbenzo[*b*]phosphindole 5-oxide has also been reported as a polar building block [63,64] with a co-polymer of carbazole and phenylbenzo[*b*]phosphindole 5-oxide achieving sacrificial hydrogen evolution rates of up to 6132 μmol h^−1^ g^−1^ (*λ* > 420 nm, 350 W Xe light source; water:methanol:triethylamine) and EQEs of 14.9% and 14.9% at 420 nm and 460 nm. Dibenzofuran has also shown potential as a polar building block. The co-polymer with dibenzo[*b*,*d*]thiophene sulfone had a lower contact angle compared to the co-polymer of dibenzo[*b*,*d*]thiophene sulfone with fluorene and a higher sacrificial hydrogen evolution rate of 147.1 μmol h^−1^ compared to 20.7 μmol h^−1^ for the fluorene analog (λ > 420 nm, 350 W Xe light source; water:methanol:triethylamine) [64]. Cyano-functional groups have also been introduced into linear conjugated polymer photocatalysts. A styrylthiophene polymer was found to have a much higher contact angle against water of 140° compared to poly[(*E*)-3-phenyl-2-(thiophen-2-yl)acrylonitrile] with 119°, poly[(*Z*)-2-phenyl-3-(thiophen-2-yl)acrylonitrile] with 74° and the double cyano-substituted poly [2-phenyl-3-(thiophen-2-yl)fumaronitrile] with 50°. This resulted in an increased sacrificial hydrogen production rate under visible light and in the presence of ascorbic acid by a factor of 31 for poly[(*Z*)-2-phenyl-3-(thiophen-2-yl)acrylonitrile] compared to the non-cyano substituted material (8.78 mmol g^−1^ h^−1^ vs. 0.28 mmol g^−1^ h^−1^). The *E*-isomeric material had a similar hydrogen evolution rate (7.60 mmol g^−1^ h^−1^) while the double substituted material was inactive despite its excellent wettability as its proton reduction driving force was lowered [83].

Small polar side-group functionalization has also been explored as a strategy for enhancing wettability. Benzodiimidazole based oligomers with varying amounts of carboxyl substituents were synthesized using suitable monomers in a condensation reaction. This led to the organic semiconductors having increased polarity and short-range crystallinity. The enhancement of wettability and photophysical processes resulted in efficient photocatalytic hydrogen evolution with a rate of 18.63 mmol g^−1^ h^−1^ (4 wt% Pt cocatalyst, 0.2 M ascorbic acid as a hole scavenger and broadband irradiation) [84].

Post-synthetic modification has also shown potential: Copolymers of 4,8-bis(5-(2-ethylhexyl)thiophen-2-yl)benzo [1,2-*b*:4,5-*b’*]dithiophene and dibenzo[*b*,*d*]thiophene sulfone building blocks were tuned by increasing the oxidation state of the thiophene units of the former to investigate the effect on photocatalytic hydrogen evolution [85]. The partially oxidized PBDTTS-1SO showed a high sacrificial hydrogen evolution rate under AM1.5g irradiation (97.12 mmol g^−1^ h^−1^) and a high EQE of 18.5% at 500 nm. While the most wettable polymer is not the best performing material, this is nevertheless an interesting approach to increase the wetting of polymer photocatalysts.

Triazines as a nitrogen-containing class of heterocycles have been extensively studied in so-called ‘triazine-based frameworks’ (CTFs) [9,29,62,86]. The materials are intrinsically more wettable due to the polarity of the triazine group and materials with very high activity have been reported. Nevertheless, the dispersibility of CTFs is still variable when studying a set of materials as the wettability is varied, which has a strong and significant impact on the photocatalytic hydrogen production efficiency. Thus, CTFs with suitable band structure and good dispersibility are more likely to achieve higher photocatalytic hydrogen production rates [9].

For studies of sacrificial hydrogen production from water, organic scavengers such as diethylamine [1], triethylamine [56], triethanolamine [2], and or co-solvents, such as methanol [56] or *N*,*N*-dimethylformamide [82] are often used. These organic solvents are fully or at least partially water miscible (in the case of triethylamine). Their presence makes the resulting mixtures more compatible with the apolar conjugated polymers and thus results in them dispersing better. When studying water oxidation typically only silver nitride is used as the electron scavenger, thus the material has to be relatively water compatible to disperse well and function as an efficient photocatalyst. It is therefore fully rational that polar materials have been reported as photocatalysts for water oxidation: Triazine-containing CTFs are sufficiently wettable to disperse in silver nitrate solution and facilitate water oxidation on their surface [21,67,87]. For example, RuO_2_-loaded CTF-T1 which has phenylene-linkers between the triazine units produces >10 µmol g^−1^ h^−1^ under visible light irradiation (λ > 420 nm, 300 W Xe light source) [62], and cobalt-loaded bipyridine containing Bpy-CTF–Co-3 had an oxygen evolution rate of approx. 250 µmol g^−1^ h^−1^ under visible light irradiation (λ > 420 nm, 300 W Xe light source) [87].

In a study of linear conjugated polymers it was also found that the most wettable and thus dispersible dibenzo[*b*,*d*]thiophene sulfone homopolymer P10 had the highest activity compared to poly(*para*)phenylene and other nitrogen-containing but less dispersible photocatalysts, with an oxygen evolution rate of 332 µmol g^−1^ h^−1^ under broadband irradiation and in the presence of silver nitride (300 W Xe light source) [88]. Aza-CMP nanosheets, which were generated through the condensation of benzenetetramine and hexaketocyclohexane were also found to be active photocatalysts for oxygen evolution with a rate of 3.3 µmol h^−1^ for the exfoliated material in the presence of silver nitride and under visible light irradiation (λ > 420 nm, 300 W Xe light source) [89].

Similarly, no organic components are present in the liquid phase when studying overall water splitting, therefore polarity is important and rationally polar units have been used in overall water splitting systems: CTF-HUST-A1-*^t^*BuOK loaded with 4.5 wt. % Ni_x_P and 3.0 wt. % Pt facilitated overall water splitting with a hydrogen evolution rate of 25.4 µmol g^−1^ h^−1^ and an oxygen evolution rate of 12.9 µmol g^−1^ h^−1^ [86]. The homopolymer of dibenzo[*b*,*d*]thiophene sulfone (P10) was able to produce stoichiometric amounts of hydrogen and oxygen in a Z-scheme with BiVO_4_ and using FeCl_3_ or FeCl_2_ as the electron mediator with rates of 5 and 2.7 μmol h^−1^ under visible light irradiation (λ > 420 nm, 300 W Xe light source) [90]. P10 was later found to drive overall water splitting as a single particulate system after loading with Ir/IrO_x_ under visible light (H_2_: 5.6 μmol h^−1^ and O_2_: 1.8 μmol h^−1^) [19]. Similarly, a derivative of P10 was found to be active for overall water splitting after loading with FeOOH as the photocatalyst with rates of approx. 3.4 μmol h^−1^ for hydrogen and 1.5 μmol h^−1^ for oxygen production [91]. A three component system consisting of two nitrogen-containing nanosheets and reduced graphene oxide has also been reported to facilitate overall water splitting with approx. rates of 10 μmol h^−1^ for hydrogen and 5 μmol h^−1^ for oxygen production visible-light irradiation [92].

### 4.2. Polymer Photocatalysts with Polar Side-Chains

The incorporation of polar building blocks, such as triazines and dibenzo[*b*,*d*]thiophene sulfone into the aromatic backbone of polymer photocatalysts does not only result in increased polarity and thus wettability of the material, but also effects the band-positions, charge-transport and other factors relevant to the photocatalytic activity [8].

Keeping the aromatic core unchanged and only modifying attached side-chains on these polymers allows better control over the materials properties while increasing wettability. This has been investigated in the literature clearly demonstrating that the extent of polarity in the sidechains of polymer photocatalysts, as well the degree of a side-chain’s solubilizing-nature, can play key roles in the context of wettability, enhanced catalytic activity and solution processability respectively [93,94,95].

Reports of solution-processible conjugated polymers for application in photocatalytic hydrogen evolution have sparked interest in terms of scalability due to the opportunities in post-synthetic film fabrication. This is enabled by the employment of side-chains and has proved a useful concept in the manufacturing of organic photovoltaic devices. Applying the same concept, involving polymer photocatalysts for application in solar-to-chemical energy conversion, has been explored and the films still display photocatalytic activity. However, initial reports showed no improvement or were even found to be detrimental to the performance. Co-polymers of phenylene and alkyl-carbazole were found to be less active photocatalysts compared to their unsubstituted insoluble polymer analog, which most likely originates from the reduced wetting of the materials [93]. The interaction of the interface between the photocatalyst surface and the aqueous medium in the context of photocatalytic water splitting can be greatly enhanced with hydrophilic side-chains: A concept with profound implications in photocatalytic water splitting.

A series of copolymers involving phenylene and diphenyl diketopyrrolopyrrole building blocks, and employing side-chains on the diphenyl diketopyrrolopyrrole units differing in chain length and hydrophilicity, were studied for hydrogen production (Figure 10) [96]. All polymers display very similar light absorption on-sets and bandgaps thus highlighting the impact of investigating wettability in isolation by side-chain engineering. Comparing alkyl polymers to the ethylene glycol polymers in both cases the more polar material showed enhanced hydrogen evolution rates with 1% Pt loading, >400 nm cutoff filter and TEOA as a hole scavenger (PDPP3B-O4 relative to PDPP3B-C4 displayed HERs of 5.53 and 2.33 mmol h^−1^ g^−1^, whilst PDPP3B-O8 relative to PDPP3B-C8 displayed HERs of 0.50 and 0.05 mmol h^−1^ g^−1^). In this study it was observed that the improved dispersibility of the polymers with hydrophilic side-chains considerably improved the photocatalytic activity.

A similar study explored non-polar fluorene co-phenylene (FP-R) and fluorene-*co*-dibenzo[*b*,*d*]thiophene sulfone polymer photocatalysts (FS-R) with different side-chains (Figure 11) [95]. The side-chains utilized were different length alkyl side-chains and triethylene glycol (TEG), which was the most hydrophilic of the series. Through modifying the polarity of the side-chains rather than the backbones, this allowed a study where the relationship between hydrophilicity and catalytic activity could be investigated independent of significant changes to the optical properties and band-positions. Considering the optical band-gaps of both the FP-R and FS-R series individually, each member of the specific series differs by no more than 0.1 eV/23 nm, when altering the side-chain, except for methyl-substituted materials [95]. In line with expectation, it was found that the incorporation of a more polar backbone unit results in enhanced photocatalytic activity, i.e., the FS-R materials are substantially more active than the FP-R materials [62,71,95]. Furthermore, the incorporation of the polar TEG sidechains enhances the polymer affinity for water as evident by their low contact angles against water (FS-TEG 69.6° and FP-TEG 72.0°). Indeed, by far the largest sacrificial hydrogen evolution rates under visible light reported in this study are in clear correlation with the most wettable polymers with a sacrificial hydrogen evolution rate of 72.50 μmol h^−1^ under visible light irradiation for FS-TEG, which is more than double the rate of every other FS-R analog. FP-TEG was found to be 30 times more active than the FP-R analogues with a rate of 7.65 μmol h^−1^. FS-TEG shows also a high EQE of 10% at 420 nm.

Interestingly, environmental atomic force microscopy and quartz crystal microbalance measurements in air and contact with water/scavenger indicate that the FS-TEG material swells in contact with water, which is not observed for the alkyl analog FS-Hex. This indicates that more catalytic sites are accessible for the FS-TEG material under the reaction conditions and potentially also explains the extended lifetime of the electron polaron formed after photoexcitation and hole scavenging. Molecular dynamic simulations corroborate the increased degree of interaction of the aqueous environment and the more hydrophilic polymers with TEG side-chains [95].

Another study was conducted utilizing oligoethylene glycol (OEG) sidechains for a different series of conjugated copolymers (Figure 12). The HER rates also corroborate the link between polymer hydrophilicity and performance enhancement, in which a 90-fold improvement is reported (using ascorbic acid and pH adjusted to 4 as the hole scavenger and a >300 nm filter in the photocatalytic experiments) [94]. In this case, on top of the enhanced polymer-water affinity, the authors also suggest an enhancement of interaction between the polymer and Pt cocatalyst particles as a significant contribution to photocatalytic activity, hence proposed charge transport mechanisms between the photocatalyst and cocatalyst are well-facilitated.

Hydrophilic side-chains of this nature have also been shown to be useful in the preparation of polymer photocatalyst nanoparticles by flash nanoprecipitation. This allowed the fabrication of nanoparticles without a surfactant thus highlighting further advantages of employing polar side-chains. The nanoparticles of the co-polymer of oligo(ethylene)carbazole and dibenzo[*b*,*d*]thiophene sulfone combine conjugated polymers with polar cores and polar polyethylene glycol side-chains giving high sacrificial photocatalytic hydrogen evolution rates up to 7.2 mmol g^−1^ h^−1^ under broadband irradiation and an EQE of 1.60% at 420 nm [97].

Beyond oligo(ethylene glycol) side-chains, further examples have emerged in the literature which have hydrophilic functional groups directly attached to the side-chains of conjugated polymers. Recently, adenine groups anchored at the end of alkyl side-chains of conjugated polymers have been reported for sacrificial photocatalytic hydrogen production [98,99]: Spirobifluorene-based conjugated polymer networks were made to induce porosity in the material; one incorporating a fluorene monomer with di-*n*-hexyl side-chains at the bridgehead carbon atom as the monomeric partner, named PF6-SF; and one incorporating a fluorine monomer with adenine groups covalently anchored to the end of the hexyl side-chains as the monomeric partner, named PF6-SFA (Figure 13).

Whilst the hexyl chains are hydrophobic in nature, the adenine groups provide a hydrophilic component which can conveniently interact with water molecules effectively thereby enhancing the interaction of the conjugated polymer with the aqueous environment. The smaller contact angle for PF6-SFA corroborates this suggestion, as well as the observed better dispersibility upon sonication in the aqueous medium. The measured bandgaps are very similar regarding PF6-SF and PF6-SFA, indicating the adenine functionality has minimal effect in the inherent bandgap of the polymer. Indeed, a remarkable increase is observed in the photocatalytic hydrogen evolution rate (17.46 mmol h^−1^ g^−1^ for PF6-SFA vs. 0.59 mmol h^−1^ g^−1^ for PF6-SF using >400 nm filter and a 1:1:3 TEA/MeOH/H_2_O solution, no addition of Pt cocatalyst) [98]. A related study incorporates the dibenzo[*b*,*d*]thiophene sulfone unit as opposed to the spirobifluorine unit to induce polarity on the aromatic core and conducts the same study, in which the polymer involving the adenine groups anchored on the side-chains results in a sacrificial photocatalytic hydrogen evolution rate of 21.93 mmol h^−1^ g^−1^ [99].

### 4.3. Polymer Photocatalysts with Polar, Non-Conjugated Segments in the Backbone

Interestingly, reports are now emerging involving the incorporation of non-conjugated, hydrophilic components in the core of conjugated polymers as an attempt to amplify the wettability and thus the hydrogen evolution rate. Initially, and perhaps intuitively, it was stipulated that maximizing the extension of the conjugation in the organic semiconductor maximizes the conductivity capability. However, the optimum conjugation length remains unclear in the literature so far, thus the design to break the conjugation was rationalized with conveniently hydrophilic segments that do not allow conjugation to extend. This strategy increases the interface between the aqueous environment and the photocatalyst, whilst maintaining localized conjugation lengths throughout the chain [100]. In this context it is important to mention that even short oligomers have been reported to have high activity for sacrificial hydrogen production [11] as well as polymers with alkyl-linkers in the backbone [70].

Using dibromo functionalized ethylene glycol and ethylene diamine chains of varying length co-monomers at varying ratios with di-*n*-octylfluorene and 5-phenylbenzo[*b*]phosphindole-5-oxide-2,7-diyl building blocks (a copolymer with a polar backbone already known to have high activity for photocatalytic hydrogen production [63]) several materials are obtained (Figure 14). The bandgaps of the material are similar, and increased wettability is inferred through contact angle measurements and molecular dynamics simulations. Indeed, an increase in the sacrificial photocatalytic hydrogen evolution rate is observed in almost every case involving the incorporation of hydrophilic non-conjugated segments. This indicates that photocatalysis is not dependent on maximally extended conjugation across the backbone, and that wettability is a key factor influencing activity. The best performing material was P-HEG-10 (the material with 10% of the dibromo HEG monomer and 90% of the dibromo-5-phenylbenzo[*b*]phosphindole-5-oxide-2,7-diyl monomer involved in the polymerization with the fluorene comonomer) with a photocatalytic HER of 6.94 mmol h^−1^ g^−1^, compared to 4.60 mmol h^−1^ g^−1^ for the PFBPO material which does not bear non-conjugated segments (1:1:1 H_2_O/MeOH/TEA solution, >380 nm irradiation and no additional Pt cocatalyst) [101]. Small-angle X-ray scattering experiments indicate that the materials not only form rod-like bundles of polymer strands that swell in contact with water, but that they are also inhibited from efficient chain packing at atomic length scale.

### 4.4. Plasma Enhancement as a Post-Synthetic Technique to Increase Wettability

Plasma enhancement can be used as a technique for increasing the wettability of conjugated polymers, especially for the purposes of photocatalytic hydrogen evolution from water [101]. The addition of functional groups on the side-chains may also be applicable as coordination sites of other molecular photocatalysts to form heterogeneous composites for example in photoelectrochemistry and photocatalysis, which is shown to have promise already albeit mainly with inorganic semiconductors [102,103,104]. As this is a post-synthetic technique, it may be a very powerful tool for many researchers in the area of organic semiconductors who wish to apply their materials to photocatalysis.

When the hydrophobic copolymer FS-5Dodec was plasma treated for 20 min an 8-fold improvement in sacrificial photocatalytic hydrogen production activity was observed to 1131.3 μmol g^−1^ h^−1^ compared to the untreated film with a rate of 135.8 μmol g^−1^ h^−1^ (Figure 15) [101]. Argon plasma was used which can potentially induce either oxygen containing species, such as H_2_O or O_2_, to react with the alkyl chains, or surface free radicals reacting with oxygen in the atmosphere after plasma treatment [105,106]. X-Ray photoelectron spectroscopy suggests the presence of new oxygen-bearing functionalities after the treatment, in line with expectations [101].

The contact angle values decrease significantly from 109° to 43° after treatment, while the optical on-set remains largely unchanged, demonstrating that the incorporation of more polar moieties on the alkyl side-chains of plasma-treated FS-5Dodec leads to much greater wettability [101]. It is likely this enhanced interaction between the polymer surface and the aqueous environment significantly contributes to the 8-fold activity enhancement, further demonstrating the importance of considering wettability as a property for photocatalysis optimization.

### 4.5. Conjugated Polyelectrolytes

Conjugated polyelectrolytes have emerged as very useful materials in the context of solubility, dispersibility and also provide a helpful strategy for designing nanoparticles for application in photocatalytic hydrogen production. These materials maintain similar optoelectronic properties to their conjugated polymer counterparts yet differ by the ionic nature of their side-chains due to strategic functional group incorporation at the side-chain terminus and subsequent counter-ion introduction (Figure 16). This also allows the material to maintain solution-processability even in aqueous solutions and therefore has wide implications in fields which involve the interface of organic conductive materials in aqueous media [107].

As one of the first examples a poly(thiophene) conjugated polyelectrolyte (CPE) material solubilized in water together with methyl viologen acting as the electron mediator, ethylenediaminetetraacetic acid as a sacrificial reagent and colloidal platinum stabilized by poly(vinyl alcohol) as a cocatalyst was reported for photocatalytic hydrogen production from water (Figure 17) [108].

A mass-normalized sacrificial hydrogen evolution rate of over 20 mmol g^−1^—relative to the poly(thiophene) photocatalyst—was determined over the first hour under visible light irradiation (>420 nm), and the absolute hydrogen evolution rate was approximately 0.4 μmol [108]. Whilst the absolute rate is low, the report highlighted the proof-of-concept in photocatalytic hydrogen production from water with soluble CPEs.

Heterogeneous systems involving the use of CPEs in hydrogen production have also been reported. The incorporation of the quaternary ammonium sidechains (PFBT-CPE) leads to a material which disperses much better in aqueous media because of the increased wettability with respect to PFBT (Figure 18) [109]. Furthermore, the charged nature of the side-chains facilitates nanoparticle to formation in aqueous media, therefore this class of material benefits from appreciable wettability/dispersibility and particle size as desirable structure-activity enhancers in solar fuel production.

All materials were originally synthesized by Suzuki-Miyaura polycondensation, and indeed PFBT-CPE displayed the greatest photocatalytic HER of 512 μmol h^−1^ g^−1^ in this study, which was 3.1 and 42.7-fold greater than PFBT and PFBT-C6, respectively (1:1:1 H_2_O/MeOH/TEA solution, >420 nm irradiation, no additional cocatalyst). It is worth noting that the authors also attribute the enhanced activity to a reduced bandgap and hence greater utilization of the solar spectrum, and improved charge separation too [109,110]. This report highlights that, in some cases, ionic side-chains can have a greater influence on bandgap positioning than non-ionic side-chains discussed in Section 4.2.

Other examples of quaternary ammonium based CPEs were synthesized by the Sonogashira cross-coupling reaction (Figure 19) [111].

All CPEs displayed activity for sacrificial photocatalytic hydrogen evolution from water, however, PFBr-PhCN displayed by far the highest absolute rate of 38.3 μmol h^−1^ (the equivalent of 15 mmol h^−1^ g^−1^ when mass normalized) with 3% Pt cocatalyst, an appreciably high rate relative to the literature on solution-processible conjugated polymers without ionic side-chains [111]. 

Water soluble CPEs with quaternary ammonium terminal groups have also been studied for photocatalytic hydrogen evolution and compared to the insoluble conjugated polymer precursor (Figure 20) [112]. The sacrificial photocatalytic hydrogen evolution rates increase 50-fold for the CPE with respect to the precursor (11.50 μmol h^−1^ and 4.6 mmol h^−1^ g^−1^ using TEOA as hole scavenger and 3% Pt loading, under broadband irradiation). The counter-ions were also found to have an effect on the performance of both cationic and anionic CPEs (NH_4_^+^ groups in the cationic case and SO_3_^−^ groups in the anionic case): In both series it was observed that smaller counter-ions result in greater photocatalytic HER activity, thus indicating the counter-ion size is a variable to consider for researchers in the field for optimizing activity. The authors also comment on better photocatalytic hydrogen evolution rates for the cationic CPEs as opposed to the anionic CPEs due to a proposed enhanced interaction with the Pt cocatalyst based on time-resolved photoluminescence kinetics [112].

Water-soluble hyperbranched polyelectrolytes have also shown promise as photocatalysts for hydrogen production from water [113]. In this structural framework (Figure 21), the conjugation is segmented yet the materials still possess semiconductive properties. The material was found to be highly dispersible in aqueous solutions thereby highlighting the benefit of the enhanced interface interaction between the material and the environment. Sacrificial photocatalytic hydrogen evolution rates of 1.08 mmol g ^−1^ h^−1^ under broadband irradiation were determined (using 0.2 M ascorbic acid adjusted to pH 4, 3 wt. % Pt) [113]. These types of frameworks are attractive because it allows the use of a large range of building blocks whilst maintaining solubility in water which increases photocatalytic activity.

Pyridinium-pended CPEs and porphyrin-based CPEs have been reported to be active for photocatalytic hydrogen evolution (Figure 22) [114,115]. The pyridinium-pended CPE produced a photocatalytic HER of 7.33 mmol g^−1^ h^−1^ (4:1 H_2_O/TEOA, 3 wt. % Pt photodeposition, λ > 300 nm), which was 63 times higher than its non-ionic precursor [114]. For the porphyrin-based CPEs, a series was developed in which the central atom (M) of the porphyrin unit was altered (including Ni, Cu, Pt and also no metal center). The authors report the only active material in the series without additional cocatalysts is the material with Pt in the core of the porphyrin, with a photocatalytic HER of 5.39 mmol g^−1^ h^−1^ reported. This rate was further boosted to 37.9 mmol g^−1^ h^−1^ when loaded with 3 wt. % Pt, a value amongst the highest obtained for CPE materials in photocatalytic hydrogen evolution (0.2 M ascorbic acid with no pH adjustment, wavelength of irradiation not specified) [115].

Solution-processable conjugated polyelectrolytes incorporating fluorene-derivatives and dibenzo[*b*,*d*]thiophene sulfone have also been reported (Figure 23) [116], allowing a comparison of photocatalytic hydrogen production activity between films and the bulk particulate material.

The CPE was synthesized via Suzuki-Miyaura polycondensation followed by addition of C_2_H_5_Br to form the ionic side-chain. The bulk material without any additional cocatalyst attained a photocatalytic HER of 0.15 mmol h^−1^ g^−1^, which incidentally was three times higher than the non-ionic precursor under the same conditions, highlighting the importance of the enhanced interface reaction between the semiconductor and the medium (7.5% TEOA in H_2_O solution, visible-light irradiation). Upon 3% Pt loading, the rate was boosted to 14.5 mmol h^−1^ g^−1^, and when a drop-casted film of the same Pt loaded material was used, an HER of 20.5 mmol g^−1^ h^−1^ was obtained [116]. On the consideration of scalability, preparing films of CPEs in a similar manner is an encouraging strategy.

In summary, the design of polar cores and polar side-chains has paved the way in elucidating the importance of wettability at the interface when applying polymeric materials to photocatalytic water splitting. In the pursuit of more active systems in application, materials going forward should incorporate polar moieties as a fundamental template in organic semiconductor design. A summary of literature examples discussed can be found in Table 1.

## 5. Increased Water Accessible Surface Areas

An important factor in water splitting is the number of accessible active sites for proton reduction and water oxidation. Hence, attempts to maximize the number of interfaces has been explored through the design of porous organic semiconductors (Figure 24). Early reports on carbon nitrides indicated that higher accessible surface areas are beneficial for the photocatalytic activity [117]. This led to the study of conjugated microporous polymers (CMPs) and covalent organic frameworks (COFs) in the literature as they allow for higher surface areas to be achieved combined with better control over the materials’ properties due to synthetic control. Similar to the case of linear conjugated polymers and conjugated polyelectrolytes discussed above, the important parameter of wettability was not considered initially as a primary structure-activity defining factor. As the literature on porous materials has evolved in the last decade, it is clear to see the progression in material design correlating wettability and photocatalytic activity.

### 5.1. Conjugated Microporous Polymers

First conceptualized and reported in 2007, CMPs have since seen many uses, with this review focusing on photocatalysis [118,119,120]. Some of the initial reports for application in photocatalytic hydrogen evolution emerged around 2015 [5,52].

Figure 25 displays some of the core CMP structures applied to photocatalytic hydrogen evolution early on. Microporosity and resulting surface areas can be tuned depending on the building blocks and linker lengths, including for example phenylene, pyrene, and spirobifluorene. Furthermore, the tuneability of absorption properties is also utilized, particularly in the case of the pyrene-phenylene CMPs through co-polymerization [5]. Whilst the photocatalytic activity of each CMP in this series was low (the best materials achieving a photocatalytic HER of under 20 μmol h^−1^, or 200 μmol h^−1^ g^−1^) most of the materials showed an improvement in rate with respect to poly(*p*-phenylene) (**P1**), with an HER of around 2 μmol h^−1^ or 20 μmol h^−1^ g^−1^, thus elucidating an important proof-of-concept in favor of porosity, structural and optoelectronic tuneability (>420 nm filter, 4:1 H_2_O/diethylamine solution, no additional cocatalyst except for PPP which was loaded with Ru) [5,52].

Whilst the number of interfaces is increased when comparing CMPs to linear analogues, the CMPs described above are still inherently apolar, thus in an aqueous reaction medium (as is the case for photocatalytic water splitting), the interaction of the semiconductor with the reactive constituents in its environment is unfavorable. In other words, the wettability of the pores in the material is low. In order to improve on the rates based on this concept, many articles have since been released with more polar constituents that increase wettability.

Heptazine-based microporous networks were reported as catalysts for sacrificial photocatalytic hydrogen evolution [123]. These materials are close analogues of carbon nitride, however, they can be obtained at much lower temperatures through condensation reactions (Figure 26). The sacrificial photocatalytic hydrogen evolution rates were approximately 2 μmol h^−1^ under >420 nm irradiation (10% TEOA as electron donor and 3% Pt cocatalyst loading), thus lower than polymeric carbon nitride under the same conditions, but outcompeting poly(*p*-phenylene) (P1) irradiated using >290 nm irradiation [123].

Doping strategies were employed to boost visible light photocatalytic activity of the HMPs. The report of donor-acceptor HMPs (in which dianiline and benzothiadiazole-dianiline building blocks are incorporated into the network) showed an increase in the photocatalytic activity in comparison to carbon nitride to a rate of over 30 μmol h^−1^ (5 wt% Pt loading, 10% TEOA in H_2_O solution, >395 nm filter) [124]. The increase was attributed to the donor-acceptor interaction which enhances the photogenerated charges, yet the potential increase in wettability was not explored.

The donor-acceptor concept was explored further with so-called sulfur- and nitrogen-containing porous polymers (SNPs) (Figure 27a). The best performing materials achieve a sacrificial hydrogen evolution rate of over 45 μmol h^−1^ (10% TEOA in H_2_O solution, 3 wt. % Pt loading, pH 7 buffer, >395 nm cutoff filter) [125]. In this work, the authors comment on wettability, in which the SNPs with the most polar structures are greater dispersed and are therefore the most active. If the SNPs of the two most active materials are broadly compared, the BET surface areas are rather similar (698 and 656 m^2^ g^−1^ respectively). Hence, it is likely that the more polar material of the two outperforms the other because of the wettability enhancement, however this is not explored further [125].

Following on from these studies, many microporous polymers have emerged as catalysts for photocatalytic hydrogen production. Amino-functionalized conjugated porous polymers, in which the amine group is incorporated onto the side-chains of a fluorene building block which is not in conjugation with the backbone (named Tr-F3N), was reported with BET surface area of 105 m^2^ g^−1^ (Figure 27b) [126]. The sacrificial photocatalytic hydrogen evolution rate reached 446 μmol h^−1^ g^−1^ (>300 nm, 1:9 TEOA/H_2_O, no additional cocatalyst), which under these conditions can be deemed, at the very least, competitive with the report of the original CMPs discussed above. Wettability for the amino-functionalized material Tr-F3N was increased as evident from contact angle measurements with values of 19.9° for Tr-F3N and 91.6° for the unfunctionalized counterpart Tr-F8, thus Tr-F3N has an enhanced interaction at the interface of the pores and the aqueous medium (for comparison, the unfunctionalized CPP had a BET surface area of 97 m^2^ g^−1^ and had a hydrogen evolution rate of 15.5 μmol h^−1^ g^−1^) [126].

Pyridyl-based CMPs produced very high photocatalytic rates (the best around 18,000 μmol h^−1^ g^−1^, 2:1 water/MeOH, 0.1 M ascorbic acid, AM 1.5 filter 380–780 nm, 2 wt. % Pt cocatalyst loading). Direct comparison to the first examples of CMPs are difficult as the materials are only tested after loading with Pt [128].

Very high photocatalytic hydrogen evolution rates without any additional Pt cocatalyst were observed for a CMP of thiophene and pyrene (CP-St, Figure 27c) synthesized by Stille cross-coupling (BET surface area of 516 m^2^ g^−1^) [55]. Using a range of different co-solvents for the photocatalytic experiments resulted in increased activity. It was found that exfoliation occurs when solvents, such as *N*,*N*’-dimethylformamide (DMF) and *N*-methyl-2-pyrrolidone (NMP) are used. This significant morphological enhancement of the interface, along with the wettability enhancement induced by the thiophene moieties, leads to a photocatalytic hydrogen evolution rate of more than 300,000 μmol h^−1^ g^−1^ being reported (>420 nm irradiation, NMP/AA/H_2_O solution adjusted to pH 4, loaded with 0.5 wt. % Pt) [55]. Under broadly speaking the same conditions except for the sacrificial reagent and co-solvent, this system is a significant improvement with respect to CMPs reported previously. The authors further demonstrate the applicability of solvent exfoliation on related CPPs producing systems with high photocatalytic HER activity [16].

More recently, cyano-functionalized CMPs have been made by Suzuki cross-coupling of cyano- and dicyano-phenylenes with pyrene units. The most wettable CMP, according to contact angle measurements, produced the highest photocatalytic hydrogen evolution rate of 10,500 μmol h^−1^ g^−1^ (10:1 H_2_O/TEOA solution, 3 wt. % Pt, >420 nm irradiation). The authors also comment on the enhanced donor-acceptor interaction contributing to the enhanced rate, highlighting the challenge of isolating one structure-activity relationship in isolation for any organic semiconductor material in this field [65].

Another example of an efficient donor-acceptor CMP based on carbazole linkers synthesized by Suzuki cross-coupling has been reported. Across a series of four CMPs, a clear trend is observed in which the most polar and therefore most wettable materials display the greatest photocatalytic hydrogen evolution rate (the highest achieving a rate of 15,300 μmol h^−1^ g^−1^, 2:1 H_2_O/MeOH, 0.2 M ascorbic acid adjusted to pH 4, >380 nm cutoff filter, no additional cocatalyst) [129]. Furthermore, it should also be noted that the most active material was by far the most porous (BET surface area of 1530 m^2^ g^−1^) as well as having the most polar constituents, which will likely contribute to the rate superiority.

Owing to the success of linear conjugated polymers incorporating the dibenzo[*b*,*d*]thiophene sulfone unit, naturally the use of incorporating this building block into CMP architecture has been explored in the literature for use in photocatalytic hydrogen production (Figure 27d) [127]. Clearly with respect to apolar CMPs, the incorporation of the sulfone unit will have the effect of increasing the wettability of the pores. With a BET surface area of 811 m^2^ g^−1^ and synthesis route via Pd-catalyzed Suzuki-Miyaura polycondensation, this material serves as a porous analog to the linear polymer containing dibenzo[*b*,*d*]thiophene sulfone. The authors report an HER of 2460 μmol h^−1^ g^−1^ (4:1 H_2_O/TEOA, >420 nm irradiation, no additional Pt cocatalyst), which outcompetes apolar CMPs and at the very least is competitive with the linear homopolymer analog (no study was done in this work to exactly confirm this under the same conditions) [127].

Meaningful cross-comparison across the literature is difficult in this field because of the lack of standardization as there are many variables that must be considered which can influence the photocatalytic activity of any system. Overcoming this issue, studies have attempted to study the influence of porosity and/or polarity by comparing materials with different levels of porosity and analogous linear polymers: A study compared linear copolymers with microporous copolymers in which a phenylene unit is replaced with a spirobifluorene unit to induce porosity; and the second comparing linear and microporous dibenzo[*b*,*d*]thiophene sulfone copolymer analogues [37,64].

In the former study, it is clear that the most polar materials produce the highest photocatalytic hydrogen evolution rates across all polymers studied, however, there is not a particularly obvious trend between analogous microporous and linear polymers, at least for this series. For the most polar materials, photocatalytic hydrogen evolution rates are similar between the porous and non-porous counterparts, yet the least polar microporous materials clearly have enhanced rates in comparison to linear counterparts [64]. The study elucidates the important concept that wettability and porosity influence activity but not necessarily harmoniously.

In the latter study, involving linear and microporous dibenzo[*b*,*d*]thiophene sulfone polymers compared to apolar related structures, the polar materials significantly outperform the apolar materials due to a large contribution from increased wettability. Secondly, the best performing CMP material, S-CMP3 (Figure 28) outperforms its linear analog P35 (>420 nm filter, 1:1:1 H_2_O/MeOH/TEA, no additional cocatalyst), highlighting the advantage of the increased interfacial sites as a consequence of the CMPs porous structure. Furthermore, S-CMP3 shows significant water uptake and shows evidence of swelling as the volume mean diameter increases significantly when comparing pure H_2_O to the H_2_O/MeOH/TEA solution. Whilst this is also observed in the linear analog, the increase is much larger for the CMP (factor of 4.4 in comparison to 1.67 for the linear analog). Therefore, it could be that the CMP has an increased swelling ability which means the catalytically active interface amongst the pores is more exposed to the reactive environment, thereby enhancing the rate of hydrogen production.

S-CMP3 shows the greatest degree of water uptake in comparison to the fluorene-analog F-CMP3 and P35, which indicates the penetration of water into the pores of S-CMP3 is the greatest as a result of the enhanced wettability of the material. It is likely that the existence of porous interfaces, the swelling ability and the water uptake capacity all coalesce to the most active material in this series. Quasi-electron neutron scattering experiments (QENS) confirm that water is taken up into the porous structure of the materials, with S-CMP3 showing higher uptake compared to the non-polar fluorene analog F-CMP3. The QENS experiments also show that water is far more mobile in the polar S-CMP3 material compared to the non-polar F-CMP3 which strongly indicates that mass transport is also more difficult under photocatalytic conditions [37].

These studies provide some valuable data on what likely occurs at the interface between the organic semiconductors and aqueous media during photocatalysis. Exploring this further, a recent study investigated H_2_O adsorption onto a porous hyper-crosslinked polymer (HCP) consisting of a triazine monomer and a biphenyl monomer [130]. A combination of density functional theory (DFT) and diffuse reflectance infrared Fourier transform spectroscopy (DRIFTS) was used to analyze potential adsorption sites and describe the adsorption of H_2_O vapor to interact with both the triazine and biphenyl unit but more strongly with the latter [130]. Whilst this material is a HCP, the principle of this process could be applied in the context of photocatalytic water splitting with conjugated materials in order to understand design principles based on favorable interactions at the interface.

Processes at the interface between CMPs, water and scavenger were also studied by neutron spectroscopy [131]. The study suggests that bound water at the interface is beneficial for photocatalytic hydrogen production, i.e., water that interacts with the surface of the photocatalyst. A higher degree of interaction at the interface between water and the S-CMP3 is inferred from the data, and it is likely the influence of the spiro unit, as well as the contribution of polarity from the sulfone groups, that result in the better interaction. Furthermore, no correlation of a strong interaction is observed regarding the interaction between the hole scavenger triethylamine and the CMP, indicating the wettability of the CMP is a strong contributor to its activity.

Polymers of intrinsic microporosity, therefore materials that have an unbranched backbone but are nevertheless porous in the solid-state, have also been shown to be active for photocatalytic hydrogen production from water. Here, materials also containing wettable sulfone-containing units show the highest photocatalytic activity [132].

CMPs have also been reported to facilitate overall water splitting in the absence of any sacrificial reagents. Whilst the process of water oxidation is extremely challenging when using organic semiconductor photocatalysts, a report has emerged detailing the use of CMP nanosheets as visible light photocatalysts for stoichiometric water splitting. 

Both materials straddle the water oxidation and reduction potentials allowing for simultaneous photocatalytic H_2_ and O_2_ production. The solar to hydrogen efficiency is reported as 0.6% with a quantum efficiency of 10% at 420 nm for PTEPB (Figure 29) [13]. This report appears particularly insightful due to the fact that no sacrificial reagents or cocatalysts are added to facilitate OWS (as well as no residual palladium present during synthesis), and what is also particularly striking about the OWS activity is the lack of wettability in this material as it is inherently apolar. The authors provide theoretical calculations as a proposition for the OWS mechanism; however this will require further investigation beyond computationally suggested intermediates. More work is needed to confirm the reproducibility of these results as there has been no follow up studies since this report to date.

### 5.2. Covalent Organic Frameworks

Covalent organic frameworks (COFs) are porous materials which have gained great interest in optoelectronics for their inherent properties, in particular for their long range order and accessibility via noble metal free synthesis [133]. Linear conjugated polymers and CMPs show little long range order, even though the transport of charges to interfaces and subsequent photocatalytic hydrogen production has been discussed as an important parameter to maximize performance [41]. First reported in 2005, the application for photocatalytic hydrogen production did not emerge in the literature until 2014 [26,134].

The COFs which initially appeared in the literature incorporated linkers such as boronate esters that are too water-labile for application in solar water splitting [135]. Water stable COFs were developed later, for example the hydrazone based TFPT-COF (Figure 30) with the crystallinity of the framework confirmed by powder X-ray diffraction. The BET surface area was determined to be 1603 m^2^ g^−1^ and the pore size was calculated to be 3.8 nm. TFPT-COF loaded with a Pt co-catalyst produced a photocatalytic hydrogen evolution rate of 1970 μmol h^−1^ g^−1^ (10% TEOA in H_2_O, >420 nm filter). However, powder X-ray diffraction measurements also found the long-range crystallinity being lost during photocatalysis and the BET surface area was reduced to 410 m^2^ g^−1^ [134]. Nevertheless, the obtained rate at the time was competitive with other organic semiconductors in terms of quantum efficiency, thus the interest in their use for solar fuel applications has grown ever since. Furthermore, approaches are being made to enhance the crystallinity of COFs in synthesis too [136]. TFPT-COF contains polar constituents such as the triazine unit, hydrazone linkers and oxygen-containing functionalities, thus it is self-evident that these materials are more polar than the initial CMPs applied to hydrogen production in the literature. The wettability of the pores will likely be enhanced, something which was initially not widely considered as a core design principle. More examples of COFs appeared in the literature for hydrogen production thereafter with imine, azine and enamine linkers obtained by Schiff-base chemistry [137,138,139,140,141].

A family of COFs with hydrazine linkers that differ by the number of nitrogen atoms in the linker unit (that is, an aryl ring with 0, 1, 2 and 3 nitrogen atoms all *meta* to each other, denoted as the N_x_-COF series) were synthesized to compare their activities for photocatalytic hydrogen production [27]. The trend was observed that more nitrogen content resulted in higher photocatalytic activity, with the highest rate of 1703 μmol h^−1^ g^−1^ being achieved for N_3_-COF (>420 nm filter, 1% TEOA in H_2_O and pH adjusted to 7, Pt cocatalyst). The authors comment that one single property in isolation was not observed for the activity differences, but rather a coalescence of factors was reasoned. This included favorable surface area, stacking capacity, and the degree of crystallinity due to the favorable dihedral angle tending towards the triazine COF for exciton mobility enhancement[27]. It is apparent though that wettability differences was unexplored for the series as a potential contribution. A report of pyrene-based COFs also emerged in the literature for hydrogen production, which produced relatively low rates with the highest achieving 98 μmol h^−1^ g^−1^ (AM 1.5 G filter, 10% TEOA in H_2_O, 8 wt. % Pt loading) [142]. This could be indicative of the poor interaction at the interface expected between an apolar photocatalyst and a polar aqueous environment, yet it is not explored further.

The importance of wettability was highlighted in the report that followed in 2018 on sulfone-containing COFs which was compared to the best member of the N_x_-COF series, N_3_-COF [143].

The COFs, named S-COF and FS-COF (Figure 31), are synthesized initially by a Schiff base reaction between 1,3,5-triformylphloroglucinol and corresponding diamino- functionalized equivalents of the sulfone monomers, followed by a tautomerization to form the enamine linker as well as the keto groups. This tautomerization provided added stability to the frameworks, crystallinity was deduced by PXRD and the BET surface areas were calculated as 985 and 1288 m^2^ g^−1^ for S-COF and FS-COF. The photocatalytic hydrogen evolution activity was reported as 4440 μmol h^−1^ g^−1^ and 10,100 μmol h^−1^ g^−1^ for S-COF and FS-COF respectively (0.1 M ascorbic acid, 8 wt. % Pt cocatalyst, >420 nm filter). On comparison with N_3_-COF under the exact same conditions (470 μmol h^−1^ g^−1^), a remarkable increase in rate is observed. The authors discuss the importance of crystallinity for the high photocatalytic activity with respect to amorphous linear polymer analogues tested under the same conditions, and the paper discusses the impact of enhanced wettability of FS-COF in comparison to N_3_-COF as well as a COF synthesized using a terphenyl unit in place of the sulfone units as an apolar COF comparison (TP-COF) (contact angles of 23.6°, 53.4° and 59.7°, respectively). Water uptake measurements are also performed on S-COF and FS-COF as a means of demonstrating the favorable interaction of water and the photocatalyst interface (67 wt. % water for FS-COF compared with 16 wt. % water for the apolar TP-COF), thereby leading to a greater rate of proton reduction [143]. Many other factors are likely responsible for the enhanced rate, including excited state lifetime, bandgap magnitude, dispersibility and the extent of porosity to name but a few, however it is also very likely that rates are further boosted by the strong wetting ability of the sulfone-containing COF materials [144].

Whilst linkers such as imine functionalities are useful functional groups, they are nevertheless not completely inert to hydrolysis especially in non-neutral pH conditions. This has strong implications in the field of solar fuel production due to the commonly used acidic and basic sacrificial reagents employed in practice [133]. Vinylene linkers have emerged recently via Knoevenagel condensation reactions to form the COF architecture, which not only results in sp^2^ connectivity between constituents but also embeds linkers which are more inert under the photocatalytic conditions to enhance the stability of the photocatalysts. Indeed, COFs synthesized in this manner have shown competitively high hydrogen evolution activity [145,146,147,148]. For instance, the sp^2^-carbon-linked COF (Figure 32) based on benzobisthiazole units has a reported HER of 15.1 mmol h^−1^ g^−1^ (0.1 M ascorbic acid, 8 wt. % Pt loading, >420 nm irradiation), although no wettability rationale is discussed [149].

More recently, a more-wettable analog has since been reported, also with benzobisthiazole constituents yet differs by the incorporation of the 1,3,5- triformylphloroglucinol building block in place of the benzotrithiophene building block to form an enamine linker, which produced an HER of 48.7 mmol h^−1^ g^−1^ (3% Pt loading, 0.1 M ascorbic acid, >420 nm irradiation) [150]. This is a rate which is competitive with the S-COF and FS-COF (Figure 31), which indicates the importance of incorporating wettability-inducing components in harmony with efficient light-absorbing components (such as dibenzothiophene sulfone and benzobisthiazole) in COFs, especially when comparing the rate obtained for the vinylene-linked analog based on the more apolar benzotrithiophene constituent (Figure 32) [143,145,150]. COFs incorporating vinylene and enamine linkers and polar comonomers will likely be explored as the field progresses.

To summarize, the linkers (Figure 33) developed in the literature for COFs have allowed highly efficient and stable photocatalysts for hydrogen generation. Linkers such as enamines and vinylenes with a cyano group are even less susceptible to hydrolysis thereby enhancing the stability of the photocatalyst. Nevertheless, all of these linkers induce an inherent degree of polarity to the structure which favors aqueous photocatalysis, something which is not attained when other organic photocatalysts are synthesized through classical cross-coupling reactions of sp^2^ carbons such as CMPs and linear polymers for instance [5]. Much of the literature broadly compares COFs to amorphous organic semiconductors stressing that crystallinity plays a major role in the activity of COF systems [143]. Through these examples of COFs with enhanced wettability, the advantage of wetting at the interface is another key advantage associated with COF structures that should feature more in the activity rationalization for photocatalytic water splitting. A summary of some materials discussed with large internal surface area are tabulated in Table 2.

## 6. Increased External Surface Area

Studies on polymer photocatalysts have shown that the size of the polymer photocatalyst particle size is an important property which can have a profound effect on photocatalytic performance. Linked to this, it is suggested in studies that the degree of particle dispersibility in the aqueous medium is in correlation with activity, in order to maximize light absorption. Factors that affect dispersibility include wettability, physical density and particle size average/distribution. A high-throughput study, intended to deconvolute common structure-activity relationships for a vast library of copolymer combinations, highlighted the importance of high dispersibility as a key factor for high activity. The best performing copolymers display an optical transmittance of 20% or less, which indicates a large degree of incident light absorption and good dispersibility is key for high activity [10].

In addition to this, it is important to consider that the relevant charge transfer events for water splitting occur at the surface-liquid interface. Thus, recombination of excitons embedded in a micrometer-sized particle will tend to outcompete the timeline necessary for excitons to be utilized for photocatalysis. This is what leads to the description of the exciton dead zone (Figure 34), where the exciton diffusion length (which is typically around 5–20 nm for conjugated polymers [151,152]) is far smaller than the distance to the surface-liquid interface [23]. However, on the nanoscale, the propensity for excitons to eventually be utilized for redox chemistry is increased, which theoretically will lead to greater photocatalytic performance with respect to bulk analogues of the same conjugated polymer. Specific literature examples on the nanoscale will be discussed.

### 6.1. Polymer Photocatalyst Nanoparticles

Perhaps the most recent consideration in the field timeline for enhancing photocatalytic activity has been to manipulate the particle size of the conjugated polymer photocatalysts. Studies in this fashion can probe direct comparisons between the bulk properties and the nanoparticle properties for any particular material as a structure-activity relationship investigation. Techniques such as mini-emulsion and nanoprecipitation synthetic strategies, and nanocomposite/nanohybrid synthetic design to name but a few, have drawn attention in recent years which has resulted in polymeric nanomaterials with promisingly high activities for photocatalytic green hydrogen evolution [23,30,33,39,116,151,152]. Exfoliation techniques in fabricating carbon nitride and CMP nanosheets, pre-organization in carbon nitride nanosheet synthesis and hydrothermal methods in carbon dot synthesis have also been utilized in nanomaterials design (refer to Figure 35 for an overview of nanoparticulate design strategies) [55,153,154,155].

A study was undertaken using a mini-emulsion polymerization technique to synthesize nanoparticle analogues of three polymer photocatalysts for hydrogen evolution. Using sodium *n*-dodecyl sulfate as the surfactant in a Suzuki-Miyaura mini-emulsion polymerization stable mini-emulsions were obtained that limit the growth of the polymer particles [156]. This strategy yields nanosized polymers that remain active photocatalysts (Figure 36).

In each nanoparticle analog, the photocatalytic activity increased by at least 2 times with respect to their bulk analogues [156]. The combination of improved catalytic lifetimes, dispersibility, and efficiency at high concentration results in an HER of 14.5 mmol h^−1^ g^−1^ for P10-e with high external quantum efficiency (EQE) of 5.8% when measured using a 1 mg mL^−1^ concentration. For comparison, this performance is much higher than the bulk analog of P10 measured under the same conditions resulting a hydrogen evolution rate of 6.13 mmol h^−1^ g^−1^ with an EQE of 2.3% (>420 nm filter, 1:1:1 H_2_O/MeOH/TEA, no Pt cocatalyst added). Furthermore, P10-e is the most wettable nanoparticle studied in the series, and the trend is observed that higher HERs are attained as wettability increases but seems not to be considered in the performance analysis in this report. This study signifies the usefulness of mini-emulsion techniques to aid the processing of otherwise insoluble bulk photocatalysts that are notoriously hard to process but are active for hydrogen production. However, it is of note that in some cases involving studies with CMPs that a significant blue-shifting was observed in the nanoparticulate analog with respect to the bulk, which indicates that achieving a high degree of polymerization may be hindered using the mini-emulsion technique in some cases [23].

### 6.2. Nanocomposites of Polymer Photocatalysts

Nanocomposites have emerged in the literature as a strategy to overcome the propensity for electron-hole recombination, by creating a heterojunction between two photocatalysts which have their bandgaps aligned in such a way to induce a type-II offset (Figure 37). This can facilitate what is known as a Z-scheme mechanism [47].

Donor-acceptor composites, such as that depicted in Figure 37, provides a driving force for exciton separation, thus allowing a more efficient use of photogenerated charges to drive photocatalysis. This, in combination with the small exciton diffusion length associated with organic semiconductors, favors utilization of the nanocomposite archetype.

A high-throughput study was conducted on 5 donor type polymers and 5 fullerene and non-fullerene acceptor molecules which conform to this archetype to form donor-acceptor nanohybrids (DANHs) via nanoprecipitation [33]. The DANHs had the effect of not only increasing the hydrogen evolution rate with regards to single organic photocatalyst nanoparticles, but also the longevity of photocatalysis increased significantly too (from 2 h degradation time to up to 18 h for DANHs) [33]. The proof-of-concept is clearly demonstrated in this work, however there was no elucidation of the most impactful structural design parameters, including hydrophilicity which this review has extensively highlighted is a key parameter to optimize in hydrogen production with organic semiconductors.

Another report documenting the concept of donor-acceptor blends was reported thereafter for a nanoparticulate system, in which a dramatic enhancement in the rate of photocatalytic hydrogen production was observed through rational design [30].

In this system, PTB7-Th acts as the donor and EH-IDTBR as an acceptor are used and the nanoparticles were synthesized via mini-emulsion with surfactants (Figure 38). Depending on the surfactant used, the morphology of the nanocomposite could be shifted from core-shell to blended. A blended morphology of donor and acceptor at the surface of the nanoparticles is not only desirable to optimize the cascade of charge separation, but has also huge implications concerning the number of charges generated at the interface of the photocatalyst and the reaction medium, a factor which greatly enables hydrogen production. Indeed, the highest activity is achieved with the optimized morphologically blended nanoparticles, with an HER of 28.1 mmol h^−1^ g^−1^ (0.2 M ascorbic acid, 5% Pt loading, >350 nm filter) [30]. This highlights the synthetic modularity available when designing nanoparticles and will undoubtedly be implemented as a necessary design strategy going forward in this area.

This study was followed up by attempting to increase the wettability of donor-acceptor nanoparticle blends through the employment of glycolated side-chains [31]. The work describes glycolated side-chains enhancing hydrophilicity and relative permittivity, which enhances the interaction at the interface as well as suppressing electron-hole recombination respectively. Water uptake and contact angle measurements corroborate the discussion in this paper, whilst previous research on high dielectric constant materials (involving conjugated polymers with polar side-chains being responsible for suppressed recombination in heterojunction solar-cells) corroborates the latter [31,158]. It is proposed that this harmonious effect in the case of glycolated nanoparticles results in enhanced photocatalytic hydrogen production, in which the best performing glycolated nanoparticle was 30-fold higher in activity with respect its alkyl-chain analog (although it is worth mentioning that the authors show the measured bandgap positioning is not identical in the alkylated and glycolated counterparts, in that the glycolated analog has a greater driving force for proton reduction by 0.3 eV which may also contribute to enhanced activity alongside enhanced hydrophilicity and relative permittivity) [31]. As such, the report convincingly shows the impact of enhancing wettability to enhance photocatalytic activity in nanocomposite systems.

### 6.3. Polymer Dots

A class of nanoparticulate material which has appeared in the literature are polymer dots (PDots), which have become very interesting materials based on the reported photocatalytic activity for hydrogen production. In 2016, Tian and co-workers synthesized organic polymer dots for this application [36].

The PDots were formed through a micellization process using the conjugated polymer (PFBT) and a polymer acting as a surfactant (PS-PEG-COOH, Figure 39). Both polymers are dissolved in THF then the solution is poured into rapidly stirring water, and as the solution is purged to exude the organic phase the PDots begin to form into a micelle conformation due to the amphiphilic nature of PS-PEG-COOH. Particle sizes were found to be in the range of 30–50 nm and the PDots exhibit a bandgap of 2.38 eV. These properties, coupled with the excellent dispersibility of the nanoparticles in the reaction medium and efficient quenching kinetics, lead to substantially high mass-normalized hydrogen evolution rates in excess of 8 mmol h^−1^ g^−1^ with ascorbic acid as a scavenger and a quantum yield of 0.5% at 445 nm. Additionally, the authors demonstrate the quantum yield could be increased 5-fold through the addition of a Pt cocatalyst [36].

Notably, they report a suitable stability of the PDots in an oxygen atmosphere which is ideal for commercial viability, and a control study was performed by synthesizing PDots without the fluorene unit which showed very little activity, highlighting the importance of planarization in materials design [36]. It is also noted that this reported mass normalized HER is 5 orders of magnitude greater than the bulk counterpart PFBT. However, it should be noted that aggregation during photocatalysis is highlighted as an issue by the authors and that the absolute amount of H_2_ produced from a system using PDots is inherently small, therefore high concentrations are required for commercial scalability. Furthermore, deactivation occurs only hours into the irradiation experiment, thus strides are required to enhance stability. Nevertheless, the concept of PDots provides a desirable architecture for the important interaction at the interface of the otherwise hydrophobic semiconductor and the aqueous environment, thus it can be envisaged that this model is applicable to even the most hydrophobic active materials in this field to boost the activity of the system.

Following this work, the same group reported even greater visible-light photocatalytic hydrogen evolution activity for PDots in 2017 [159].

PDot 2 (involving PFODTBT with the PS-PEG-COOH, Figure 40) showed a high HER of 50 mmol g^−1^ h^−1^, with an AQY of 0.6% at 550 nm. The authors identify through density functional theory (DFT) the importance of the benzothiadiazole unit for harvesting the photogenerated electron density as a result of PDot light excitation which results in efficient proton reduction [159]. It is also noteworthy that the authors probe the potential influence of residual Pd from synthesis playing a role in proton reduction, in which Pd poisoning experiments were conducted which infers that the PDots are actually responsible for photocatalysis in these studies.

More recently, a fully panchromatic system involving ternary PDots was reported utilizing PFBT, PFODTBT, PS-PEG-COOH and a small molecule acceptor (ITIC, Figure 41) to induce a heterojunction and efficient solar energy harvesting, which in harmony with particle size considerations result in efficient sacrificial photocatalytic hydrogen evolution with a Pt cocatalyst [160].

Due to the narrow HOMO-LUMO gap associated with ITIC it allows light harvesting in lower energy region of visible light, thereby creating a panchromatic system [160]. Rates of up to 60 mmol g^−1^ h^−1^ were achieved with an EQE of 7.1% reported at 600 nm, indicating the efficiency of light harvesting for this system. Furthermore, the authors report good stability of the system which is essential for scalability [160].

Other strategies in this area have led to appreciable results, including the use of different surfactants, morphology control to obtain more hollow PDots, modulation of the hydrophobic and hydrophilic constituents to optimize the micelle morphology, and even their use in biohybrid systems owing to their tunable surface morphology and pH stability [161,162,163,164]. Nevertheless, in general these examples of using PDots for photocatalytic HER emphasize the photophysical processes as the significant contributing factors to enhanced activity. However, it is also worth noting another enhanced-activity-trend in the literature involving these materials: That they outperform their bulk analogues [36]. This clearly emphasizes the impact that reducing the particle size to the nanoscale has on activity, especially considering the typical exciton diffusion length for organic semiconductors discussed previous [23]. Furthermore, what perhaps is not appreciated enough is the wettability advantage of PDot architecture, a concept which is very likely to contribute to the high rates achieved for PDots reported in this field. Work is needed to improve on stability, nevertheless these results are encouraging owing to the reported activity in the literature.

### 6.4. Polymer Nanosheets

The production of thin 2D nanosheets of photocatalytic semiconductors via exfoliation has been shown to be a highly productive strategy for enhanced performance in solar fuel production. These materials benefit from extended planarity to aid conductivity, high surface areas to maximize catalytically active sites at the interface in contact with the medium, and optoelectronic tuneability to name but a few [110].

Reports of carbon nitride nanosheets emerged first in the field, with many reports that develop activity enhancement through heteroatom doping and defect engineering for instance [150,163,164]. Progress has now been made using conjugated polymer based nanosheets, and promising results have been reported [165]. As an example, in 2020 porous polymer nanosheets prepared through liquid exfoliation in *N*-methyl-2-pyrrolidone were reported. The nanosheets displayed a HER of 190.7 mmol h^−1^ g^−1^ with ascorbic acid as a hole scavenger, which greatly outperformed the bulk analog (Figure 42) and further benefited from desirable porosity as a technique to enhance hydrogen production activity [55]. This exfoliation technique has been followed up since to design more active materials, and more will undoubtedly emerge to create a large family of materials fabricated in this manner owing to the high activities reported because of the optimal interaction at the interface in photocatalysis [16].

Furthermore, overall water splitting has also been reported using conjugated polymer nanosheets with appropriate co-catalysts that drive water oxidation [13,98,166]. Since the examples of photocatalytic overall water splitting involving organic semiconductors are few and far between in the literature to date, the fact that two reports involve nanosheet assembly clearly emphasizes the importance of this architecture as a promising target for interface optimization in the pursuit of maximally sustainable solar fuel production going forward, in which no sacrificial electron donors are necessary.

To summarize, when designing conjugated materials for applications in many areas of photocatalysis in the future, the particle size consideration should be of significant importance. Not only can the exciton utilization propensity increase, but also the dispersibility in the aqueous medium can increase too by operating the system on the nanoscale with respect to bulk analogues. Importantly, wettability can still be enhanced at the nanoscale by the strategic structural design through polar moieties to optimize the processes occurring at the interface of the organic semiconductor with the aqueous surroundings. Table 3 provides a summary of some examples discussed with small particle size and thus large external surface area.

## 7. Interface between Polymer and Metal Co-Catalyst

As discussed in Section 2 typically a metal-catalyst is needed to facilitate the charge transfer of electrons and holes from the semiconductor to protons and/or water [45,46]. Efficient interfacial charge transport between the semiconductor and the co-catalyst is not only required for high activity of the system but will also improve the stability under operating conditions as the formation of reactive intermediates is suppressed.

By far the most used co-catalyst for hydrogen evolution is platinum [2,166]. In the case of conjugated polymer photocatalysts, metals are often already present within the material originating from the catalyst used in the material synthesis, such as residual palladium that was used in the Suzuki-Miyaura cross-coupling reaction which acts as the co-catalyst for proton reduction [17,37]. Under cross-coupling reaction conditions the catalyst resting state is a Pd(0) species. Aggregation involving several Pd(0) centers can occur during the reaction, which then form palladium nanoparticles that precipitate out of solution to form palladium black [167]. Metals can also be loaded onto the materials using impregnation methods [168], and photocatalytic deposition [2]. It is important to note here though that photocatalytic deposition can give significant variations in loadings, particle sizes, and distribution of different metals depending on the sacrificial reagent used, which will affect the performance of the photocatalyst [169]. Due to its sluggish kinetics involving a four-electron transfer process, the water oxidation half-reaction remains very challenging for organic photocatalysts. Metal-catalysts, such as cobalt [88], iridium [19], NiP_x_ [86], and FeOOH [91] have been shown to facilitate water oxidation in conjunction with organic polymers.

The role of the metal-catalyst is predominantly to lower the activation the energy or over-potential at the surface of the polymer to facilitate proton reduction or water oxidation. Generally speaking, at least two steps are expected to take place that require the involvement of the metal-catalyst: An electrochemical discharge step leading to the formation of adsorbed hydrogen, followed by a catalytic step [170] that produces H_2_ via a Volmer–Heyrovský or Tafel mechanism [171]. Other mechanisms have also been suggested, for example a mechanism in which the metal facilitates the recombination of atomic hydrogen formed on the semiconductor in the case of a gold-TiO_2_ system [172]. The efficiency of this chemical process depends on the performance of the metal catalyst and is therefore not discussed in more detail here.

Another important role of metal catalysts is the separation of photogenerated excitons and suppression of recombination of charge carriers by trapping charges. For example, it has been shown that photogenerated excitons generated by poly(9,9-di-*n*-octylfluorene-*alt*-benzothiadiazole) (F8BT) undergo quenching on the femto-nanosecond time scale by palladium clusters present in the material through energy and electron transfer (Figure 43) [18]. The study also showed that palladium clusters accumulate electrons, which also increases the lifetime of charges due to suppressed bimolecular recombination kinetics. On even longer times scales these electrons are then most likely transferred to produce hydrogen. In the case of P10 (homopolymer of dibenzo[*b*,*d*]thiophene sulfone), charge separation appears to be driven by reductive quenching via a sacrificial hole scavenger in the solution phase [48]. Transient absorption spectroscopy on P10 together with spectroelectrochemical measurements strongly indicate that the polaron remains centered on the polymer and that the material struggles to transfer polarons to palladium. Nevertheless, the ability of the material to generate very high populations of long-lived electron polarons results in high activity [18], but also indicates that further development of co-catalysts will increase the activity, potentially very significantly.

The observation of the differences in the time scale of electron transfer to Pd clusters translates into different optimum concentrations of co-catalysts for maximum activity in their hydrogen evolution rates. Typically, a threshold for the co-catalyst concentration is observed: Below the threshold a reduction in activity is observed, while for higher concentrations no significant change is observed in a plateau region. At much higher concentrations, a reduction is the result of competitive light absorption of the metal on the surface [37,144,172]. In the case of F8BT an activity plateau is reached at ∼100 ppm, while at least 2 orders of magnitude higher Pd concentrations are required for the observation of an activity plateau in P10 [18].

The interface between polymer photocatalysts and the metal particles has not been well explored. As Pd(0) can have strong π-interactions with aryl rings [173] palladium nanoparticles bind well to conjugated polymers [174], which is then not removed during conventional workup [17,175]. Work on microporous poly(aryleneethynylene) materials has shown that palladium can be dispersed uniformly within these materials [176] and that palladium species can interact with alkyne-bonds [177]. X-Ray photoelectron spectroscopy studies have also shown that the nitrogen-atom in polyaniline coordinates palladium [178] and pyrimidine has been suggested to interact with platinum in a carbon nitride photocatalyst [179]. TEM images of materials that contain palladium from their synthesis, or are loaded via photodeposition, show large metal clusters which are often not particularly uniform in size [5,53]. Photodeposition also appears to be sensitive to the conditions used in the process. For a COF it was found that when ascorbic acid was used as the scavenger, platinum particles with an average size of 2.5 nm formed while basic TEOA resulted in the formation of much larger platinum particles (15 nm). Furthermore, the Pt^0^/Pt^2+^ ratio was found to be higher when ascorbic acid was used compared to TEOA [148].

X-Ray absorption spectroscopy (XAS) showed that most palladium clusters in P10 are present as Pd(0) [18]. Furthermore, a coordination number of 3.3 was determined for as-synthesized P10, which was found to be lower than the coordination number in P10 following further photodeposition of palladium (7.7) or in that of palladium foil (8.7). The Pd–Pd bond distance in as-synthesized P10 was determined to be 2.55 Å, which is longer than that of P10 following further photodeposition of palladium (2.52 Å) or in palladium foil (2.45 Å), which implies a lower degree of aggregation of palladium in as-synthesized P10. Taken together, the data suggests that small palladium clusters are present throughout the material, which are not resolved by TEM.

In the case of F8BT smaller photocatalyst particles show enhanced exciton quenching, suggesting a higher yield of electrons on palladium. This potentially indicates that palladium clusters are located predominantly in the near-surface region of these F8BT nanoparticles, which could be explained by the fact that palladium is more hydrophobic than the polymer, thus resulting in phase separation during fabrication [18]. In the case of ternary PDots (PFBT-PFODTBT-ITIC), it was found that the platinum co-catalyst acts as an additional electron acceptor [160]. The presence of platinum enhanced charge separation and prevented recombination between oxidized donor polymer (PFODTBT^+^) and the reduced acceptor (ITIC˙^−^). No direct charge transfer from the donor polymers (PFBT, PFODTBT) to the co-catalyst was observed, which only appeared to take place from the reduced acceptor that is present on the surface, thus, indicating that platinum was also deposited on the surface similar to the palladium found in F8BT. The importance of controlling the interface with the co-catalyst has been further demonstrated by a study of the hydrophobic polymer photocatalyst PFTBTA. Nanoparticles with the amphiphilic polymer PS-PEG-COOH show poor photocatalytic performance as the platinum co-catalyst appears to be interacting with the nonconjugated PS-PEG-COOH on the surface of the nanoparticles. Nanoparticles that use conjugated electrolytes allow for Förster resonance energy transfer-type electron transfer to take place and fast electron transfer from the conjugated electrolyte to platinum, which results in a significant increase in photocatalytic activity [163].

Potential electron transfer sites between carbon nitride and the metal-cocatalyst have been suggested. 2,5-Dibromo pyrimidine functionalized carbon nitride has shown significantly enhanced hydrogen evolution rates and the pyrimidine-sites were suggested to be the site of charge transfer from the material to platinum based on TA experiments [179]. Cyano-groups have also been suggested to bind to palladium single atoms in engineered carbon nitrides based on experimental observation and theoretical calculations [180]. Cobalt atoms can also be coordinated within the plane of carbon nitride through multiple nitrogen interactions as evidenced by a range of measurements, including extended X-ray absorption fine structure spectroscopy to explore the local environment [181].

For a series of six linear fluorene-type materials with increasing sulfone content it was found that the kinetics of electron extraction depends on the number of sulfone units to the ratio of photogenerated electrons generated in the material. A higher ratio results in a lower electron population on the photocatalyst and thus more efficient charge extraction. The study suggests that the sulfone unit acts as the electron transfer site for the electron polaron to the palladium co-catalyst, and that hydrogen evolution appears to be limited by the photogeneration rate of electrons rather than their extraction from the polymer to the co-catalyst [67].

A β-ketoenamine-linked covalent organic framework modified with platinum showed high activity for photocatalytic hydrogen production from water. The study found that platinum single-atoms are highly dispersed within the material and that a six-coordinated C_3_N–Pt–Cl_2_ species is formed that prevents aggregation. The increased activity compared to platinum clusters was ascribed to enhanced exciton migration and separation, as well as a reduction of the energy barrier for the formation of H* on the interface of the active platinum species and the COF [182].

Cobalt loaded polymeric perylene diimide (PDI) was found to have a high activity for photocatalytic water oxidation. Two different cobalt species were identified, CoO_x_ clusters and cobalt-atoms. The oxidation state of the CoO_x_ clusters in Co-PDI was determined to be +2.24, indicating that they were mainly comprised of CoO and Co_3_O_4_ rather than metallic Co. FT-EXAFS fits suggest that cobalt atoms are coordinated by six O atoms, which is a saturated coordination structure of the cobalt center and suggests that the atoms are in between PDI layers. The study suggests that the cobalt atoms act as the electron mediators connecting adjacent PDI layers while the CoO_x_ clusters act as sites for water oxidation [183].

The incorporation of chelating functional groups, such as bipyridine in the polymer backbone to coordinate metal ions [184], molecular metal catalysts [185,186], and the synthesis of conjugated porphyrin polymers [25] has also shown promise by controlling the interface with the metal-catalyst, however, this also requires additional synthetic steps.

Interfaces also play an important role in composites of conjugated polymers with inorganic photocatalysts, however, are beyond the scope of the review and we refer to other existing reviews [187].

## 8. Conclusions

For the design of materials that can be efficient in photocatalytic water splitting, it is clear that underlying processes have to be better understood. Some of the factors are already well understood, e.g., light absorption to match the solar spectrum at ground level, transport of charges to active sites, and sufficient thermodynamic driving-force for proton reduction and water oxidation, and these have to be considered in other material classes as well.

Nevertheless, studying new materials is elucidating new factors that have to be considered: In the case of organic photocatalysts the interface with water and wettability had not been widely identified to be an important factor and thus not been taken into account in materials’ design. On a macroscopic level, hydrophobicity leads to poor dispersion in water and breaking up of large particles during the sample preparation process. This results in material not being kept in sufficient quantity in the light path, thus reducing the materials’ apparent activity. The interaction on the polymer surface is also fundamentally important as water and protons have to reach the surface and active sites to undergo water oxidation and proton reduction reactions. If the local concentration is too low, then this could also negatively affect the performance.

The use of organic scavengers allows the access to a range of different electrochemical oxidation potentials [49], however their impact on dispersion and dielectric constant of the interface also needs to be considered [95]. Generally speaking, the use of organic scavengers will be aiding the dispersion of polymeric organic photocatalysts, but these can also lead to enrichment at the interface with the water/scavenger mixture to the point where only a few water molecules are able to access the surface [67]. Inert solvents can also be used as co-solvents that do not participate in the photocatalytic reaction itself but simply aid dispersion of the photocatalyst and wetting of the surface. Going forward, it will be important to avoid additional solvents from a scale up point of view. Organic components, both scavenger and co-solvents, in the aqueous phase will potentially increase the dielectric constant at the interface, thus aiding exciton separation, but also reduce the overpotential for the oxidation half reaction [95]. These effects will not play a role in overall water splitting systems, though the reduced dispersibility in pure water can at least partially explain the generally speaking lower rates that are observed for polymeric systems that have been studied for sacrificial activity, as well as overall water splitting in the absence of scavengers [19].

Particle size and extrinsic surface areas have been clearly shown to be important and beneficial. As such, these should always be considered as a potential factor when exploring structure-property relationships for materials based on their composition, i.e., if interesting trends are observed within a material series then particle sizes should be studied to rule out that their variation simply explains the difference in activity. The exfoliation of materials has also been a very successful approach and here significant potential for exploration remains.

## 9. Outlook

### 9.1. Materials

The inclusion of polar building blocks remains an excellent strategy to improve various factors in photocatalytic hydrogen production, including wettability. Given the large number of potential building blocks and potential for use in more complex materials, e.g., as statistical co-polymers [5], a large space remains to be studied. In the case of bulk photocatalysts that are dispersed in water (rather than making polymer nanoparticles) the use of polar side-chains has resulted in significant enhancement of photocatalytic performance. Here also the chemical space has not been well explored, we expect that new polar groups will be identified and that significant gains are still to be realized given that most polymer photocatalysts are apolar materials in the polymer backbone.

In principle, predictions have shown that a large number conjugated polymers are thermodynamically able to perform overall water splitting [188] and more studies need to focus on the design of materials that facilitate this to drive this area of research forward. Related to this, it will be important to study metal cocatalysts for overall water splitting and we expect that significant performance gains are possible based on what has been observed for inorganic semiconductor photocatalysts [189]. In the longer term it will be desirable to replace these noble metal catalysts with cheaper and more abundant co-catalysts.

Understanding and optimizing the interface between the metal catalysts and polymers will also be of importance to not only increase the activity but also the stability, as catalyst detachment has been an issue in inorganic overall water splitting systems [190]. Direct coordination of metals using functional groups or attachment of molecular catalysts could potentially reduce the risk of detachment and thus improve stability of the system, but are yet to be explored.

### 9.2. Quantification

Many reports quote headline values expressed as mass normalized rates to compare to the existing literature. Others have shown that this is not necessarily a suitable method to estimate the relative performance of photocatalysts as sample mass, reactor setup, light source type and intensity, filters and other parameters will significantly affect the photocatalytic rates [191]. As such, caution must be taken when comparing different datasets as even the same material will give different gas production rates in different laboratories. For a more meaningful cross-comparison going forward, it would be highly desirable to standardize photocatalytic measurements and provide accredited testing protocols, however, the community is still far away from achieving this [192]. Experimental procedures should therefore be as detailed as possible to allow others to understand the measurement conditions (including light sources, set-up, and experimental details such as pressure). EQE values should be provided for comparison with other literature examples as these measurements remove differences in light intensity as a factor, even though other factors will be relevant that affect EQE measurements.

Finally, the use of mass normalized rates also needs to be carefully considered. In many reports the rate of hydrogen production is normalized per time unit and mass of catalyst (mol h^−1^ g^−1^). The issue with this is that the activity of a photocatalyst does not follow a strict linear relationship with mass (Figure 44a). At lower concentrations a linear relationship is observed but with higher mass no further increase is observed and potentially even reduced once going to very high concentrations due to increased scattering.

For optimum performance materials should be therefore measured in this saturation regime as the highest absolute amounts of hydrogen are produced. Based on the example in Figure 44a, photocatalyst A would be the most active material followed by photocatalyst B and photocatalyst C, in which the latter two show very similar absolute performance. When normalizing the rates depending on the amount of photocatalyst used, a different picture emerges (Figure 44b). At point X photocatalyst B has the highest mass normalized rate, and it shows the same rate as photocatalyst A at point Y. Only when going to point Z the normalization does not skew the relative order of photocatalytic activity compared to the absolute rate. It is therefore important to rule out that a normalization effect is solely responsible for the increased apparent activity.

### 9.3. Measurements

Internal surface areas are expected to be beneficial for the photocatalytic activity of these materials. The reason for this expectation is that an increased number of interfaces also increases the number of active sites for water splitting. Polarity is again important for water to wet the internal surface in these materials given that it will be confined in relatively small spaces as the pore sizes are typically in the low nanometer range. In the case of CMPs these materials are often relatively non-polar, while COFs are in many cases more polar given the polar building blocks used in the condensation reaction. However, few studies show water uptake measurements, and it is unclear if water as well as scavengers are actually penetrating into the materials. The simulation of water dynamics [193]. Overhauser dynamic nuclear polarization NMR relaxometry [194], and vibrational spectroscopy [195] could provide crucial insight in understanding water dynamics within the materials. Beyond this, the mass transport of scavengers (if required), but also desorption of gaseous products and scavenger oxidation products within these materials needs to be studied to be able to clearly understand if porosity is actually beneficial for the activity under operating conditions.

On paper, COFs have potentially a significant advantage here as their pore structure is far more defined than compared to CMPs, which in principle will allow for the design of adequate pore sizes to maximize mass transport. However, care needs to be taken when making and analyzing these materials as intrinsic disorder can be overlooked [196]. Methods that provide high quality COFs have been developed [197,198,199], but have rarely been used in photocatalysis [28]. Longer-term stability also has to be a goal as many COFs show loss of order over time under photocatalytic conditions.

New strategies to enhance the interactions at the interface have emerged through post synthetic modification methods, but we also expect that other material classes such as polymer hydrogels, hypercrosslinked polymers as well as composites with components that enhance the contact with water, and dyes for sensitization of polymer photocatalysts will be reported. Designing materials that swell in contact with water offer significant potential as evident by studies using triethylene glycol side-chains on linear polymers, and CMP materials that swell in contact with water/scavenger mixtures. A secondary effect here is also potentially that charges will be isolated by preventing charge hopping, thus extending the lifetime at the interface.

Overall, more studies are needed to generate understanding of interfaces and how other factors such as crystallinity interplay with the interfaces. Changes to local pH values and dielectric constant have also not been studied in much detail so far. The role of other interfaces also needs to be explored in more detail, for example the polymer/co-catalyst interface. Given the importance of charge transfer events, it could be anticipated that huge gains are possible here if adequately controlled.

### 9.4. Sustainability

Once materials have been identified with potential it will also be of upmost importance to ensure that the systems for photocatalytic water splitting are overall sustainable. The identification of potential issues early on will allow that these can be mitigated to minimize impact on the environment, humans, and our society.

An advantage often cited in the context of conjugated polymers is the suggestion that these materials are made from *Earth-abundant* elements [43]. While this is true in the most literal meaning as the main constituents of conjugated polymers (carbon, hydrogen) are ubiquitous on our planet, it is important to acknowledge that the monomers used in the synthesis are based on petrochemical feedstocks. Only few examples of conjugated building blocks that are obtained from renewable resources exist, none produced at scale and in most cases still requiring functionalization with functional groups that are not necessarily renewable and/or sustainable [200]. Thus, the sustainable resourcing of the starting materials will have to be considered when moving towards scale fabrication.

Another challenge will be scale up of synthesis and production itself from a sustainability standpoint: The synthesis of the photocatalysts requires several synthetic steps producing significant amounts of waste. This can be quantified in the so-called E factor, which is expressed as the amount of organic waste over amount of product. Typically, organic semiconductors will have E factors in excess of 10^4^ while commercial drug industry products with a similar number of synthetic steps have values in the 10^2^–10^3^ range [201]. Additionally, the nature of the waste has to be considered with coupling reactions such as the Stille coupling producing highly problematic organo-tin waste as expressed in their typically high environmental quotient, which is multiplied with the E factor taking the toxicity, ease of recycling, and other factors into account [202]. More sustainable catalytic reactions producing less waste, such as direct arylation reactions [203], and the use of water as the reaction solvent for polycondensation reactions have been studied [204], but more work is needed in this area. Similarly, the manufacturing process of the devices that are used needs to be considered: Using benign solvents or solvent-free methods have to be developed for fabrication at scale [205], but also substrates, glass windows, seals, and other components will significantly contribute to the footprint of these solar water splitting devices, though more sustainable alternatives are being explored [206].

End-of-life analysis will also have to play an important role in the overall assessment of device potential which will have to be performed. Recycling of the substrates and other components would be preferred, and studies are needed to explore if photocatalysts can also be reused and/or reactivated. No toxicological analysis for polymer photocatalysts exists, however, they are not expected to be problematic given that related materials, such as poly(3,4-dethylenedioxythiophene) polystyrene sulfonate have shown no cytotoxic or inflammatory response in mouse and human cells, as well as in rats [207]. Polymeric materials are also apolar and large in size which reduces the uptake by cells. However, the potential formation and escape of nano and micro-sized plastics from the active components of the devices has to be studied. Nanosized materials, such as buckminsterfullerene has been shown to possess considerable toxicological potential, though this is dependent on its surface configuration [208]. Polymer microplastics are small enough to be taken up by the biota and adsorb pollutants on their surfaces, thus enriching them on these particles [209] and the contamination of terrestrial plants potentially leads to trophic chain contamination. Furthermore, it is rational in organic semiconductors to engineer polycyclic aromatic hydrocarbons into the backbone as these fused aromatic constituents can possess useful low-energy visible-light absorbing properties, however they are linked to carcinogenicity, genotoxicity and are prone to bioaccumulation [210]. As such, device fabrication as well as waste management should be carefully implemented to minimize exposure risk.

Impact studies will play an essential role of the analysis before fabrication at scale, given that photocatalytic water splitting devices will be required at very large scale to fulfil the energy needs of our societies.

## Figures and Tables

**Figure 1 nanomaterials-12-04299-f001:**
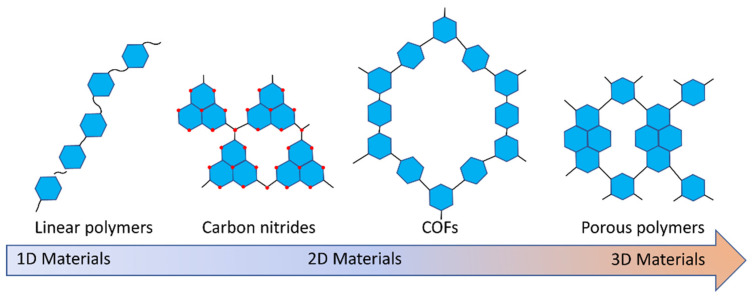
Classes of organic photocatalysts for solar fuels production.

**Figure 2 nanomaterials-12-04299-f002:**
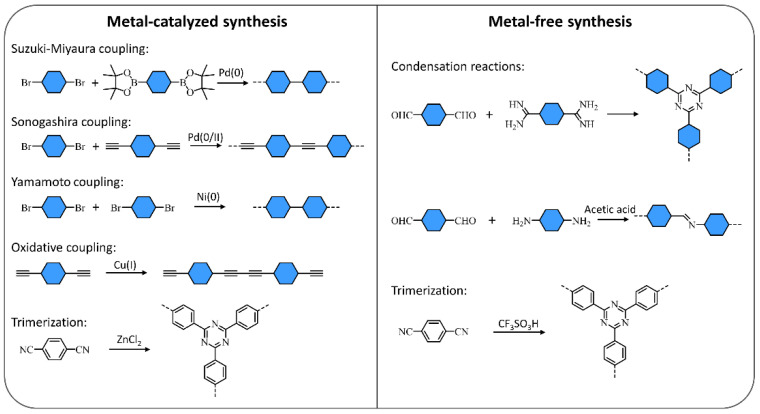
Metal-catalyzed reactions and metal-free methods that have been used to synthesize organic photocatalysts for solar fuels production [8,23,24,25].

**Figure 3 nanomaterials-12-04299-f003:**
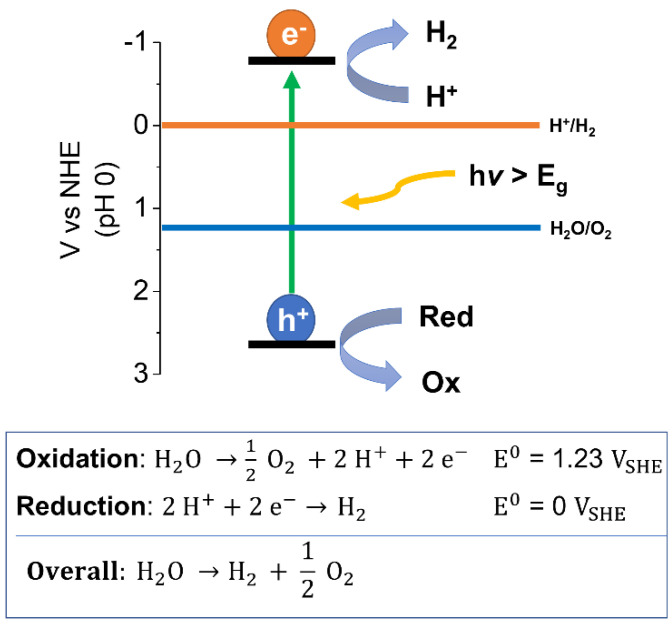
Mechanism and redox reaction involved in photocatalytic water splitting.

**Figure 4 nanomaterials-12-04299-f004:**
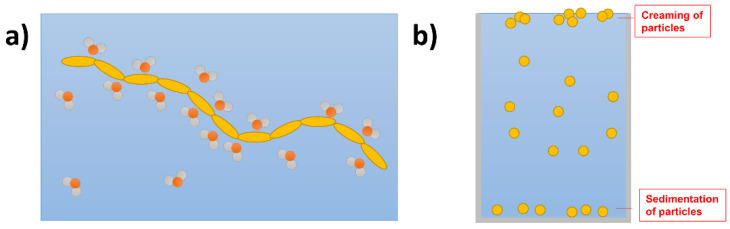
Interfaces in photocatalytic water splitting. (**a**) Interface of water with polymers in contact with water; (**b**) Dispersion of polymer particles, 100 nm–10 µm in size, in water. Creaming and sedimentation are undesired as they reduce the activity of the photocatalyst.

**Figure 5 nanomaterials-12-04299-f005:**
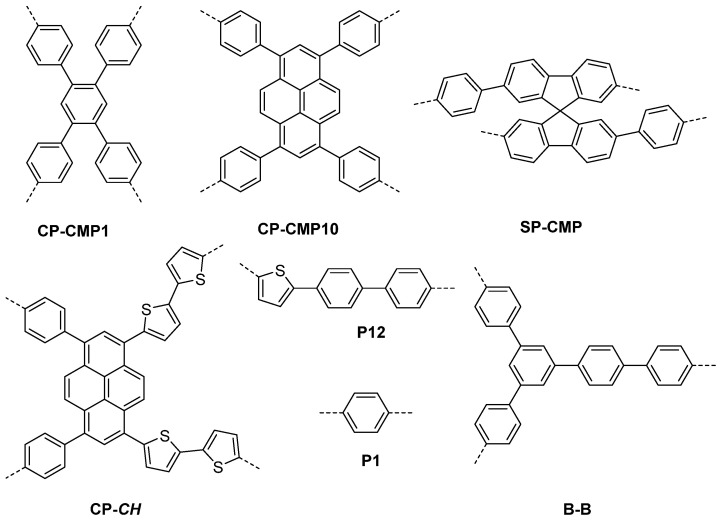
Examples of apolar polymer photocatalysts [1,5,15,16,52,55].

**Figure 6 nanomaterials-12-04299-f006:**
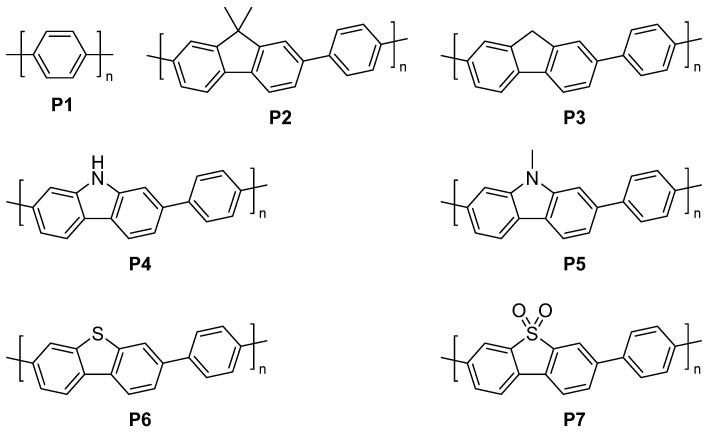
First examples of polar polymer photocatalysts [56].

**Figure 7 nanomaterials-12-04299-f007:**
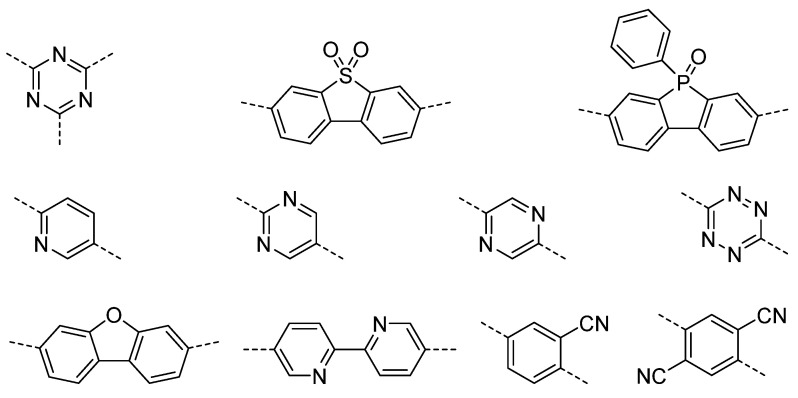
Examples of polar building blocks used to change the polarity of the backbone of polymer photocatalysts [19,32,60,62,63,64,65].

**Figure 8 nanomaterials-12-04299-f008:**
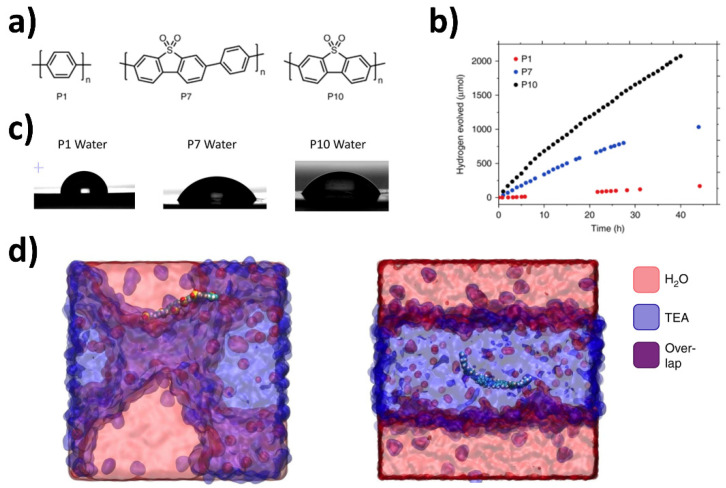
(**a**) Structures of P1, P7, and P10; (**b**) Photocatalytic hydrogen evolution experiments in water/methanol/triethylamine under visible light irradiation; (**c**) Contact angles against water for pellets of the polymers; (**d**) Snapshots of atomistic molecular dynamics simulations of oligomers of a the polar polymer P10 (left) and a non-polar fluorene polymer, as a model for P1, (right) both in a mixture of TEA (blue) and water (red) [Figure adapted from ref. [48] under CC-BY 4.0 International License].

**Figure 9 nanomaterials-12-04299-f009:**
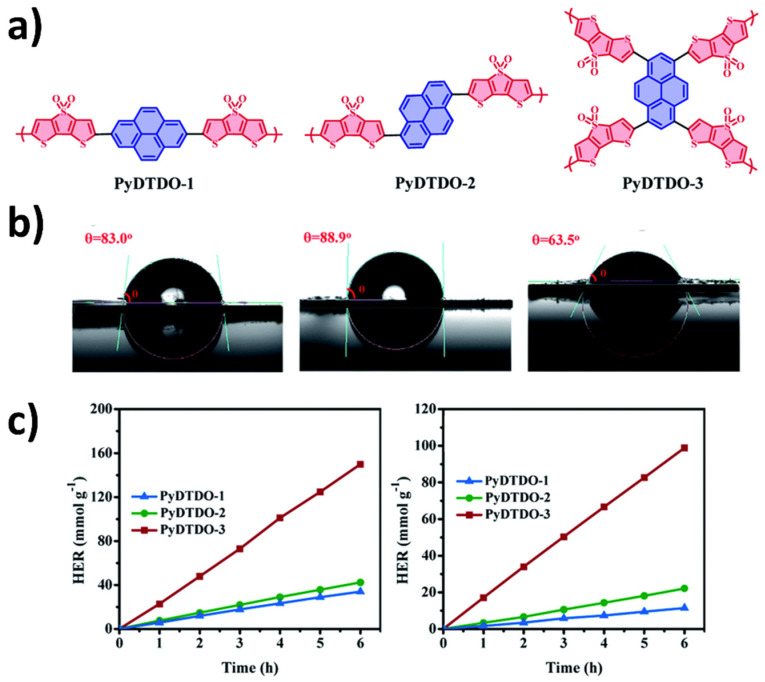
(**a**) Structures of PyDTDO-1, PyDTDO-2, and PyDTDO-3; (**b**) Contact angles of PyDTDO-1, PyDTDO-2, and PyDTDO-3 against water; (**c**) Sacrificial photocatalytic hydrogen evolution of the polymer photocatalysts broadband irradiation (left, λ > 300 nm), and visible light irradiation (right, λ > 420 nm) [Figure adapted from ref. [82] under CC-BY 3.0 International License].

**Figure 10 nanomaterials-12-04299-f010:**
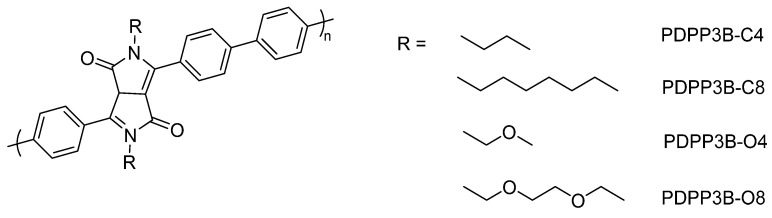
Co-polymers of phenylene and diphenyl diketopyrrolopyrrole with different side-chains to investigate their effect on the materials’ wettability and influence on their performance for sacrificial photocatalytic hydrogen generation from water [96].

**Figure 11 nanomaterials-12-04299-f011:**
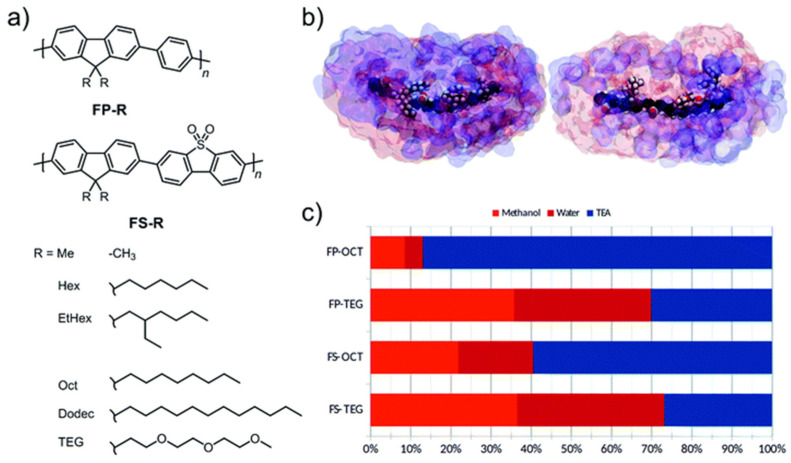
(**a**) Structures of the fluorene co-phenylene (FP-R) and fluorene-*co*-dibenzo[*b*,*d*]thiophene sulfone (FS-R) polymers; (**b**) Molecular dynamics simulations of FS-Oct and FS-TEG in 1:1:1 water/methanol/TEA. TEA is shown in blue, and the water/methanol phase is shown in pink; (**c**) Bar chart showing fraction of volume within a 2 nm radius of the oligomer that is occupied by each component in the liquid medium: methanol (orange), water (red) and TEA (blue) [Figure adapted from ref. [95] under CC-BY 3.0 International License].

**Figure 12 nanomaterials-12-04299-f012:**
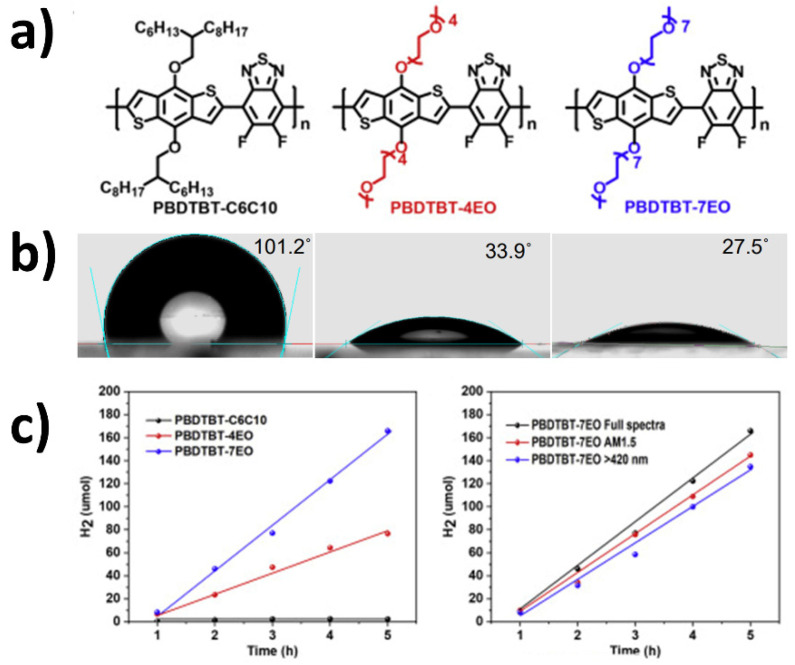
(**a**) Structures of PBDTBT-C6C10, PBDTBT-4EO, and PBDTBT-7EO; (**b**) Contact angles against water for PBDTBT-C6C10, PBDTBT-4EO, and PBDTBT-7EO; (**c**) Sacrificial hydrogen evolution rates of the conjugated polymers under broadband irradiation (left) and sacrificial hydrogen evolution rates of PBDTBT-7EO under broadband > 420 nm, and AM1.5 irradiation (right). [Figure adapted from ref. [94] under CC-BY 4.0 License].

**Figure 13 nanomaterials-12-04299-f013:**
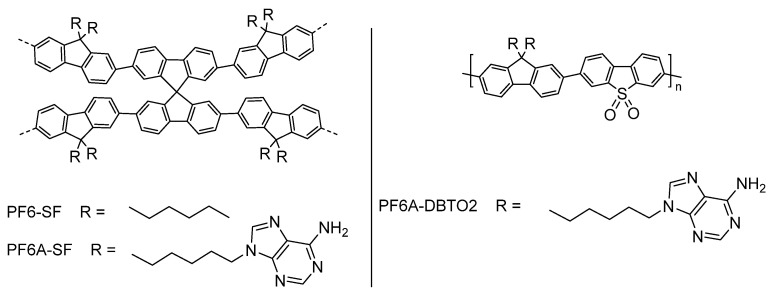
Structures of adenine substituted conjugated polymers for sacrificial hydrogen production from water [98,99].

**Figure 14 nanomaterials-12-04299-f014:**
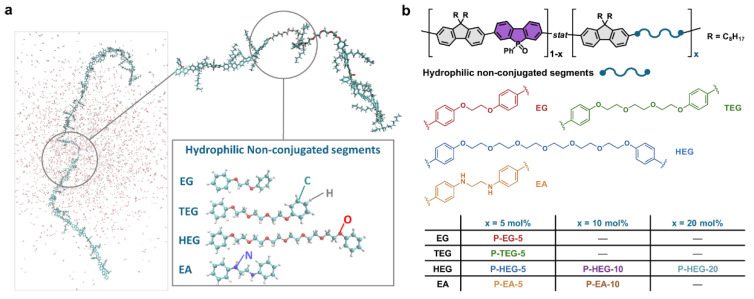
(**a**) Schematic illustration of the polymer photocatalysts containing hydrophilic non-conjugated segments; (**b**) Structures of hydrophilic segments and polymer repeat units [Figure adapted from ref. [100] under CC-BY 4.0 License].

**Figure 15 nanomaterials-12-04299-f015:**
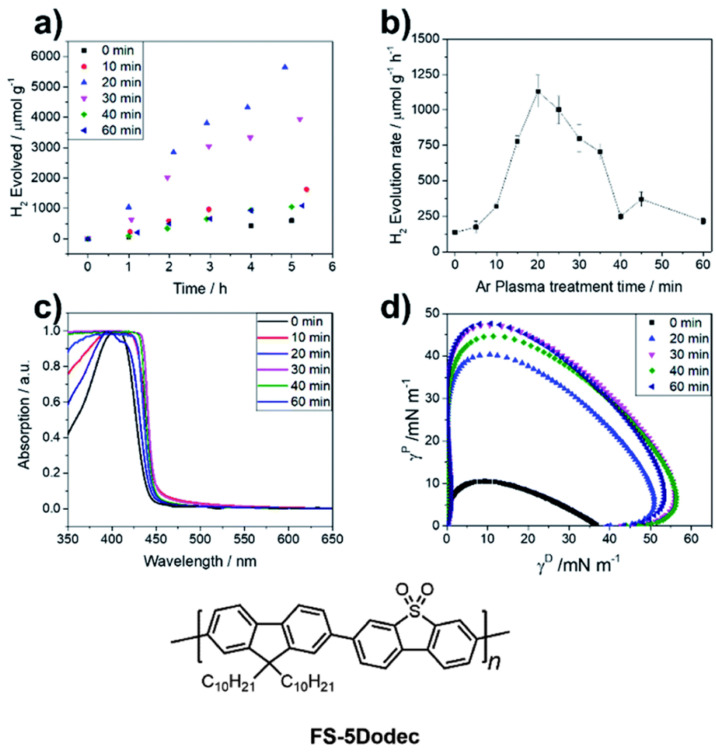
Plasma treated FS-5Dodec (**a**) Hydrogen evolution performance under visible light. Conditions: Drop-cast films on glass, 5 vol% triethylamine in water, *λ* > 420 nm, 300 W Xe light-source; (**b**) Correlation of the plasma treatment time and the hydrogen evolution rate under visible light; (**c**) UV-Visible absorption spectra; (**d**) Wetting envelopes of the polymer films. Below: Structure of FS-5Dodec [Adapted figure from ref. [101]].

**Figure 16 nanomaterials-12-04299-f016:**
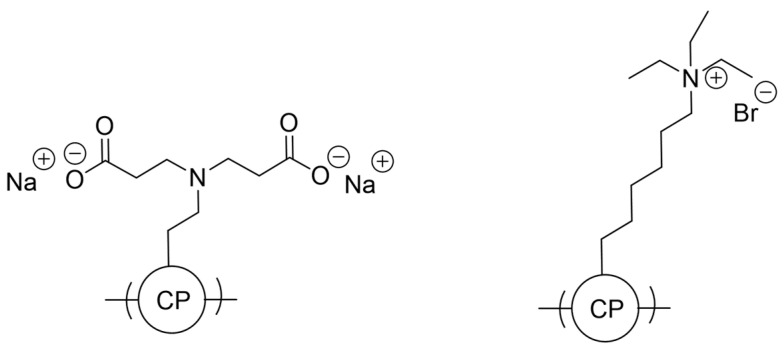
Structures of anionic and cationic conjugated polyelectrolytes to illustrate ionic component of the side-chains (CP = conjugated polymer) [107].

**Figure 17 nanomaterials-12-04299-f017:**
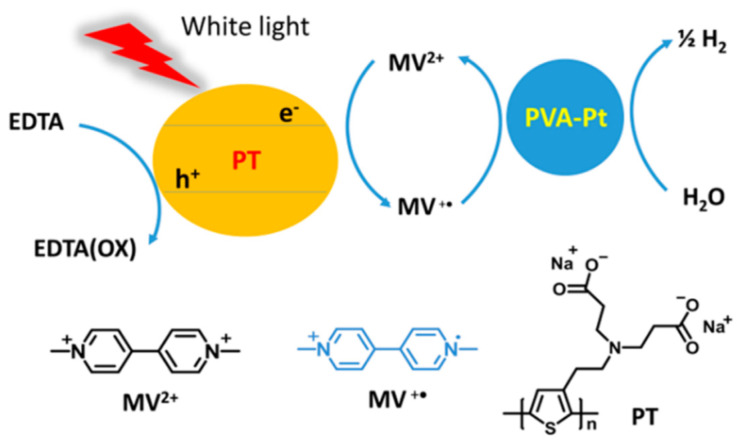
Photocatalytic system using anionic poly(thiophene), methyl viologen, and PVA-platinum for sacrificial hydrogen production from water. Reprinted with permission from [108]. Copyright 2017 American Chemical Society.

**Figure 18 nanomaterials-12-04299-f018:**
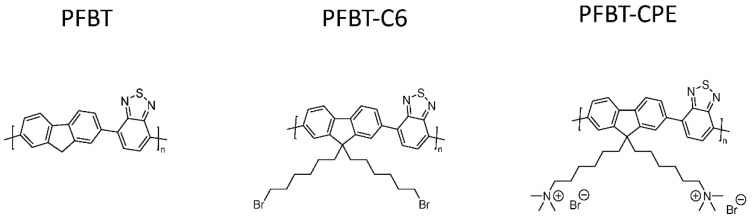
Structure of **PFBT**, precursor **PFBT-C6** and the polyelectrolytes **PFBT-CPE** [109].

**Figure 19 nanomaterials-12-04299-f019:**
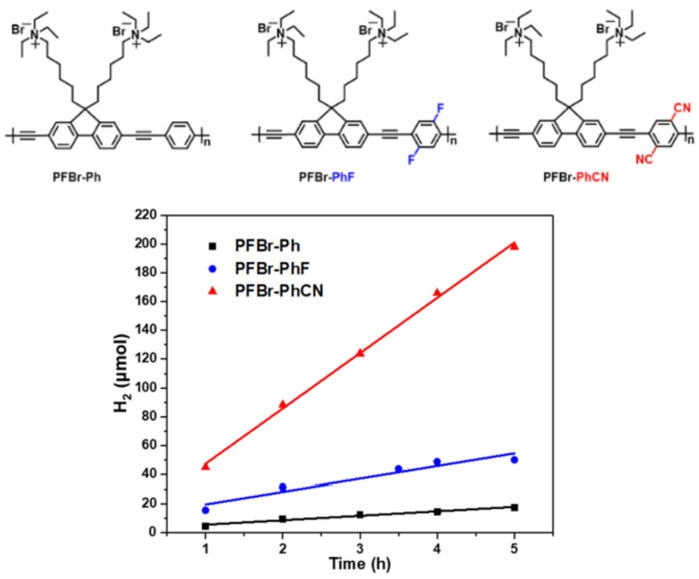
**Top**: Structures of three CPEs; **Bottom**: The corresponding hydrogen evolution data (0.2 M ascorbic acid, pH 4, 3% Pt cocatalyst, under broadband irradiation) [Adapted with permission from [111]. Copyright 2019 American Chemical Society].

**Figure 20 nanomaterials-12-04299-f020:**
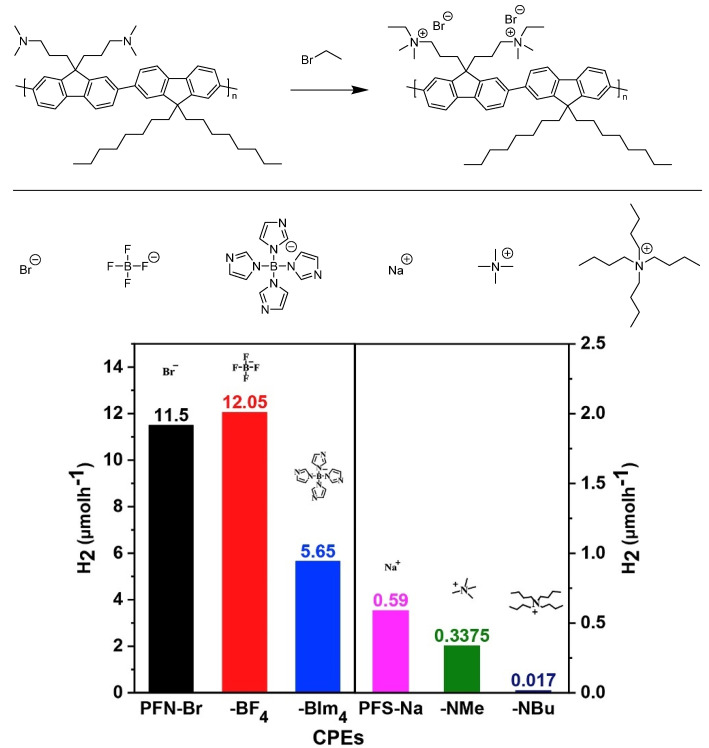
**Top**: Synthesis of a poly(fluorene) CPE from the non-ionic precursor and counter ions used in the study. **Bottom**: Rates of photocatalytic sacrificial hydrogen production for CPEs with different anionic and cationic counterions [Reprinted from Nano Energy, 60, Z. Hu et al., Highly efficient photocatalytic hydrogen evolution from water-soluble conjugated polyelectrolytes, 775–783, Copyright 2019, with permission from Elsevier] [112].

**Figure 21 nanomaterials-12-04299-f021:**
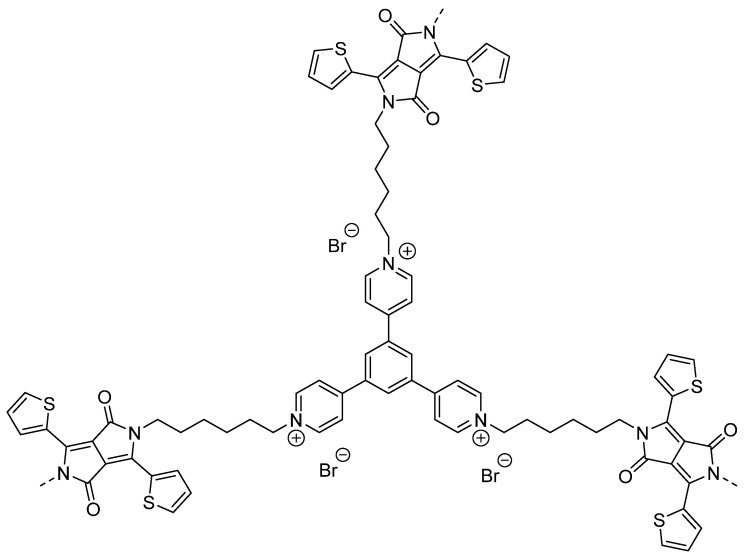
Structure of the hyperbranched polyelectrolyte active for sacrificial photocatalytic hydrogen production [113].

**Figure 22 nanomaterials-12-04299-f022:**
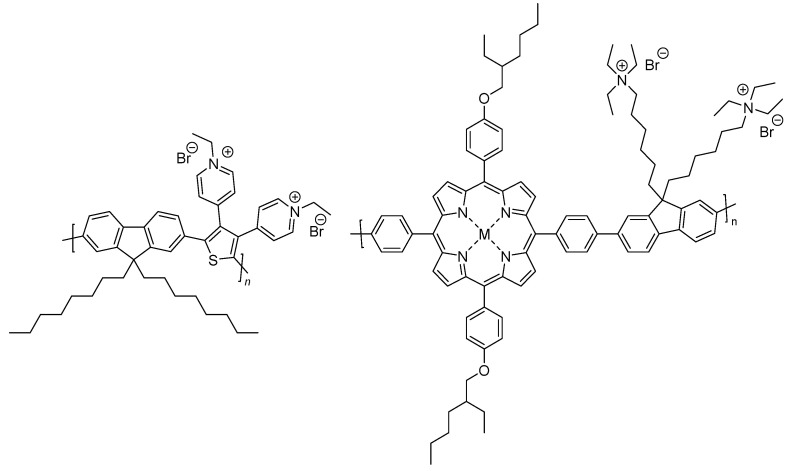
Representative structures of the pyridinium-pended CPE and the porphyrin-based CPE for application in photocatalytic hydrogen production [M = none, Ni, Cu, Pt] [111,112].

**Figure 23 nanomaterials-12-04299-f023:**
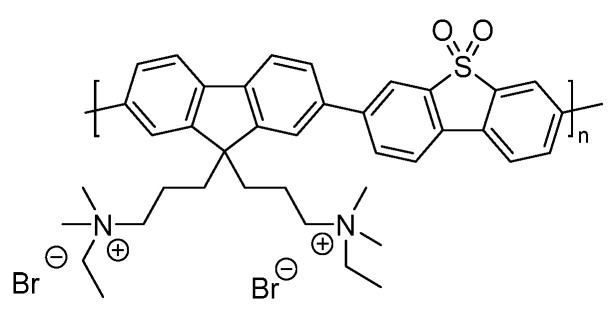
Structure of the conjugated polyelectrolyte incorporating the dibenzo[*b*,*d*]thiophene sulfone [116].

**Figure 24 nanomaterials-12-04299-f024:**
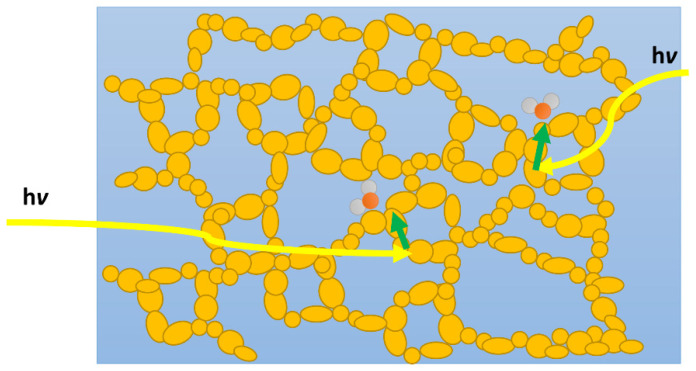
The internal surface area of porous materials potentially provides an interface for the photocatalyst and the aqueous environment. Porous materials have increased interfaces, thus no dead volume and a larger number of accessible active sites.

**Figure 25 nanomaterials-12-04299-f025:**
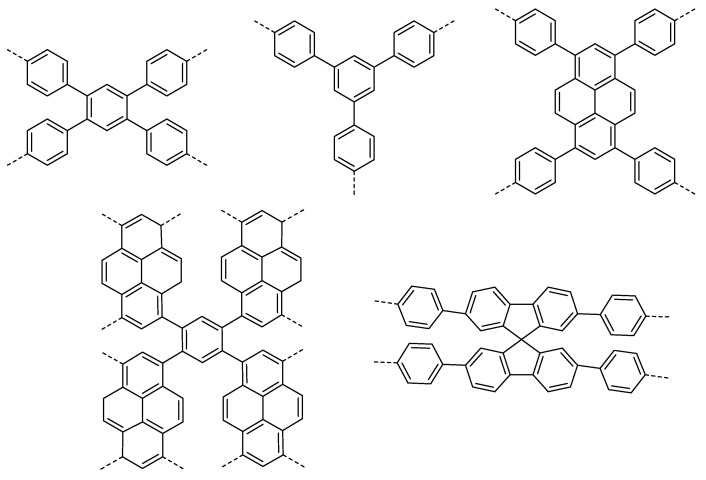
CMP structures first utilized for photocatalytic hydrogen production. Please note that the structures are much more complex than these simple representations due to their 3D nature as well as a large number of defects and end-groups [121,122].

**Figure 26 nanomaterials-12-04299-f026:**
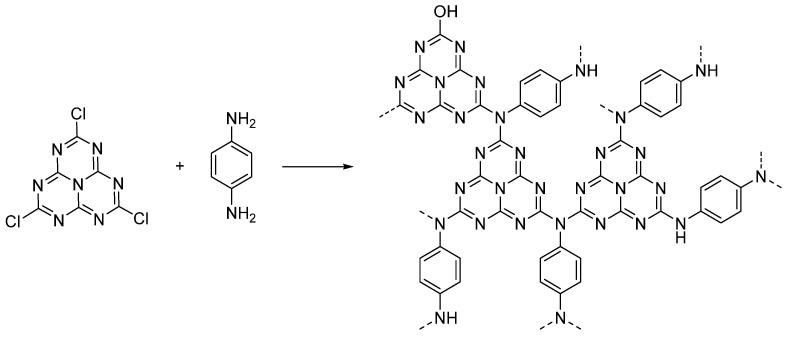
Heptazine-based microporous network obtained through low temperature condensation [123].

**Figure 27 nanomaterials-12-04299-f027:**
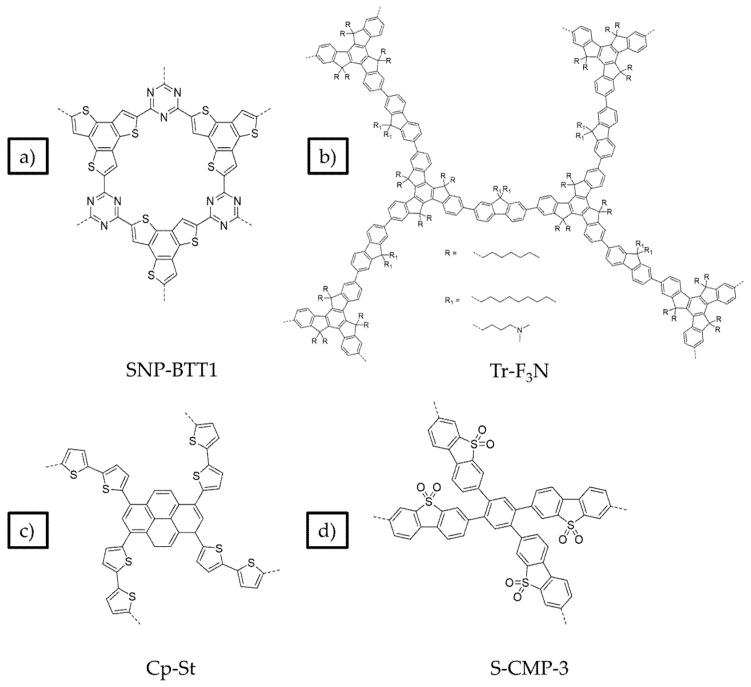
(**a**) Structure of the best performing SNPs for photocatalytic hydrogen production from water [125]. (**b**) General structure of the conjugated porous polymers designed with amino groups at the end of the chain as R_1_ (Tr-F3N) and octyl chains as R_1_ (Tr-F8) to compare each in photocatalytic hydrogen evolution [126]. (**c**) A pyrene-thiophene CMP, CP-St**,** for photocatalytic hydrogen production [55]. (**d**) Structure of a CMP incorporating phenylene and dibenzo[*b*,*d*]thiophene sulfone building blocks for photocatalytic hydrogen production [127].

**Figure 28 nanomaterials-12-04299-f028:**
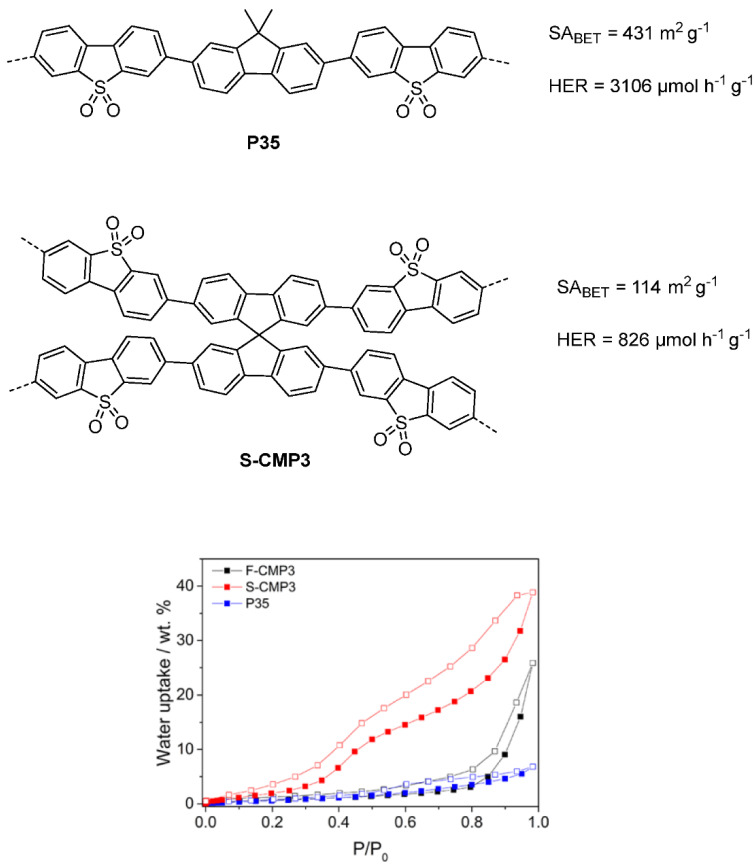
Top: Linear and microporous conjugated polymers with their corresponding BET surface areas and hydrogen evolution rates. Bottom: Water uptake measurements performed at 20 °C for S-CMP3, F-CMP3, and P35, respectively. Solid squares indicate adsorption whilst open squares indicate desorption [Figure from ref. [37] under CC-BY License].

**Figure 29 nanomaterials-12-04299-f029:**
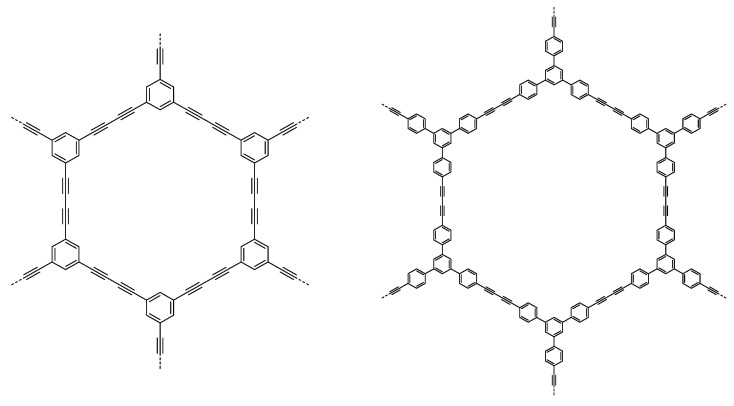
Structures of PTEB and PTEPB CMPs [13].

**Figure 30 nanomaterials-12-04299-f030:**
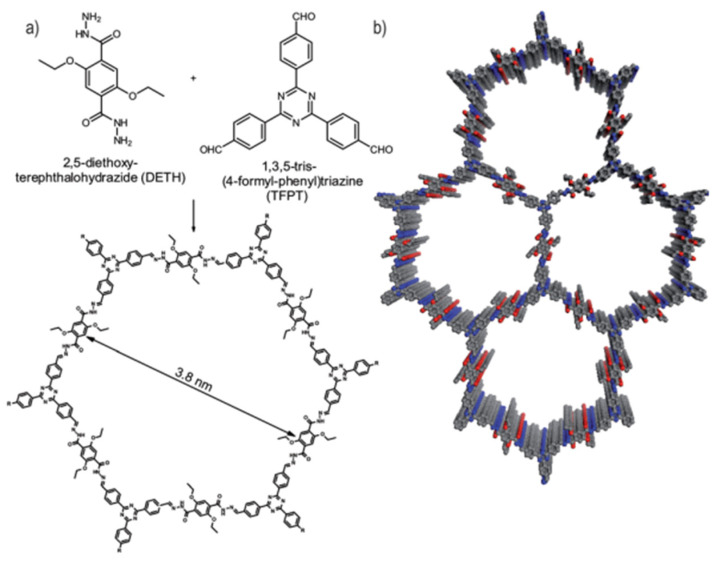
(**a**) Monomers and structure of a hydrazone-based COF (TFPT-COF); (**b**) The hexagonal lattice structure of **TFPT-COF** (grey: carbon, blue: nitrogen, red: oxygen). [Figure from ref. [134] under CC-BY 3.0 International License].

**Figure 31 nanomaterials-12-04299-f031:**
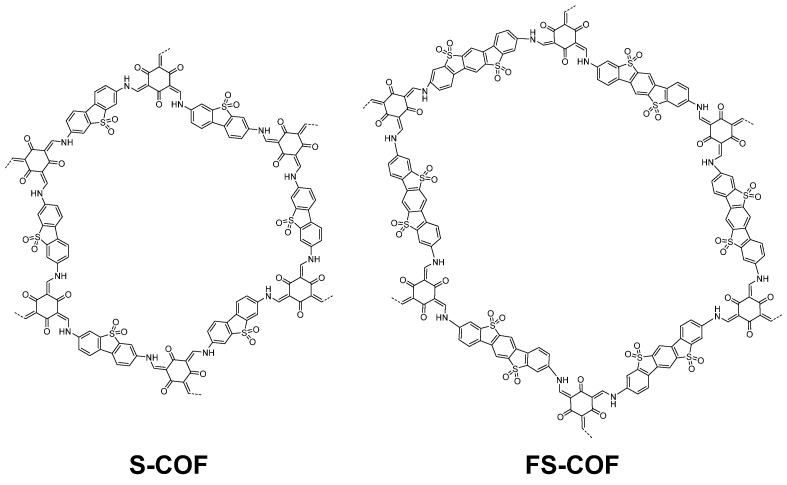
Sulfone-containing COFs, S-COF, and FS-COF structure [143].

**Figure 32 nanomaterials-12-04299-f032:**
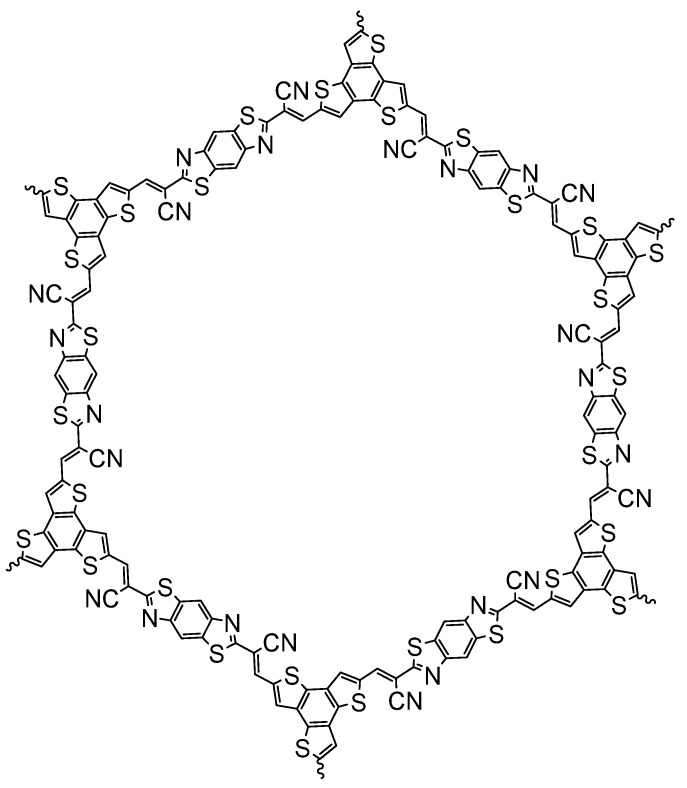
An example of a vinylene-linked COF for photocatalytic hydrogen production from water [149].

**Figure 33 nanomaterials-12-04299-f033:**
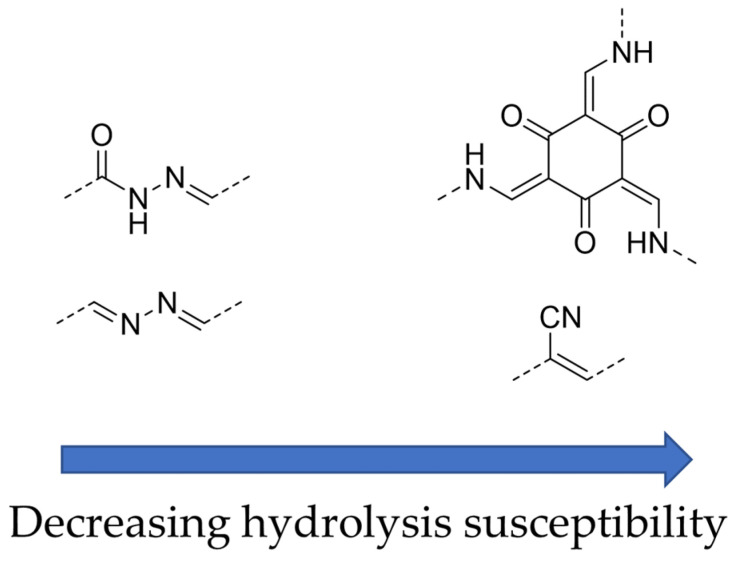
Structures of the linkers reported for COF materials active for photocatalytic hydrogen evolution with arranged according to their relative hydrolysis susceptibility.

**Figure 34 nanomaterials-12-04299-f034:**
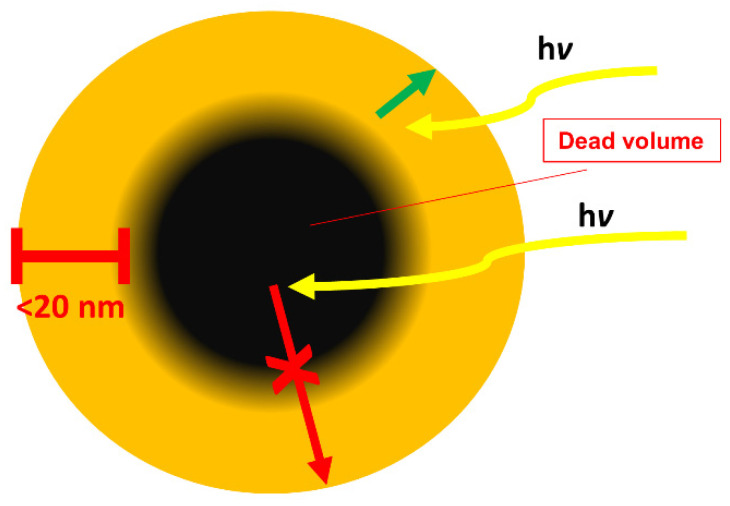
Dead volume within a polymer photocatalyst particle. Exciton diffusion lengths are limited to <20 nm which means that those created deeper within the particle are not able to reach the surface and thus do not contribute to the materials’ activity.

**Figure 35 nanomaterials-12-04299-f035:**
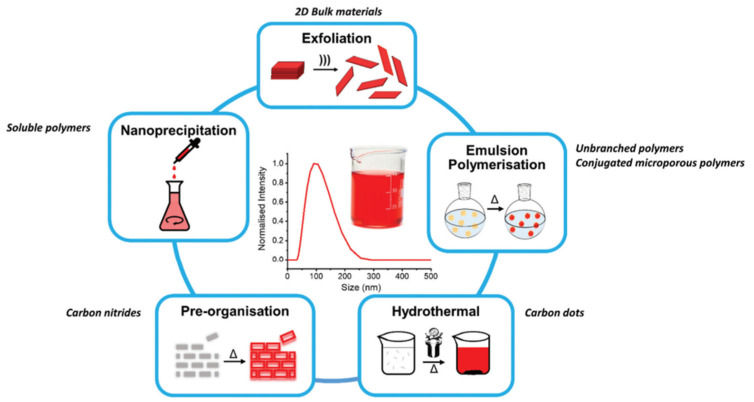
Illustration of the design strategies for conjugated polymer nanomaterials in the context of photocatalytic hydrogen production [Figure from ref. [23]].

**Figure 36 nanomaterials-12-04299-f036:**
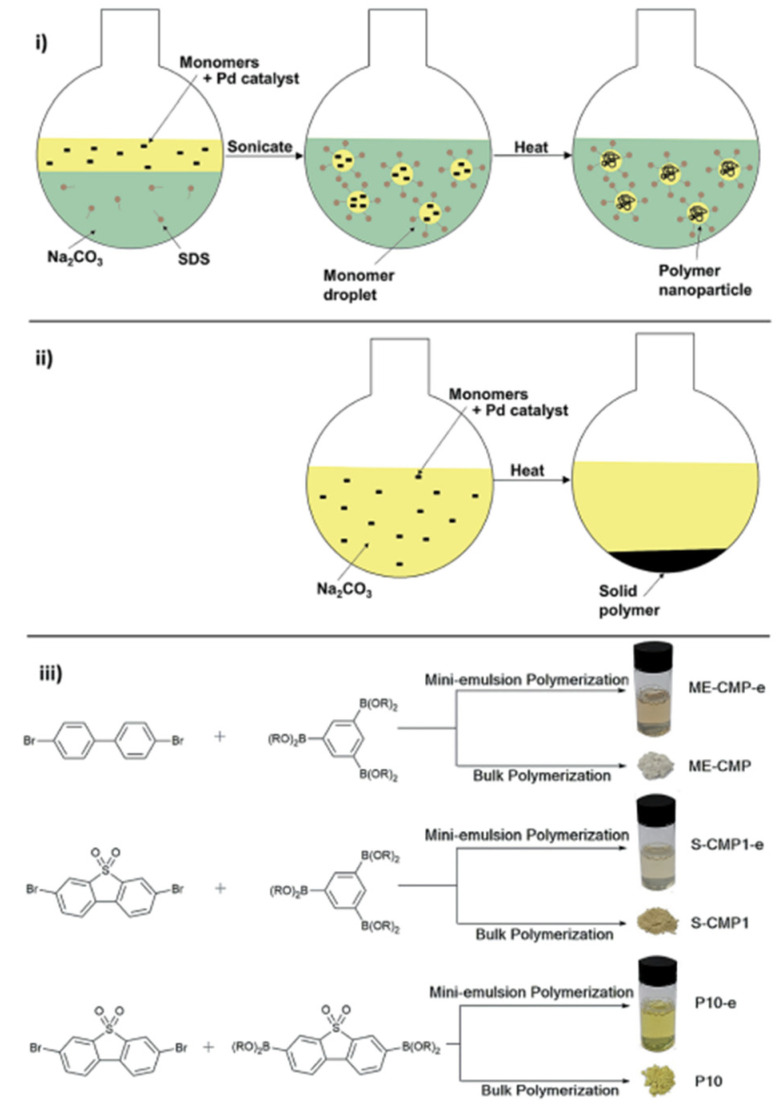
(**i**) Mini-emulsion polymerization illustration. (**ii**) Bulk polymerization illustration. (**iii**) Structures of monomers and pictures of products studied to compare the bulk HER to the nanoparticle HER [Figure from ref. [156] under CC-BY 3.0 International License].

**Figure 37 nanomaterials-12-04299-f037:**
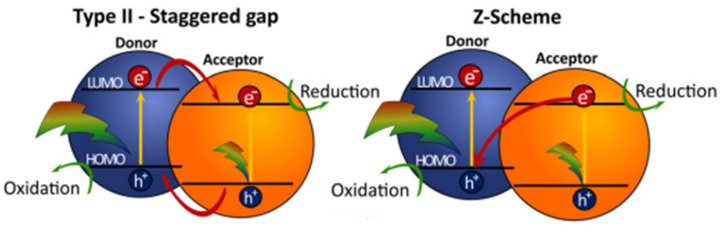
General schematic of a type-II offset to create a heterojunction in a nanocomposite photocatalyst system for hydrogen production. The charge transfer processes are also illustrated that are typical of a Z-scheme photocatalytic system. [Figure from ref. [157] under CC-BY 3.0 International License].

**Figure 38 nanomaterials-12-04299-f038:**
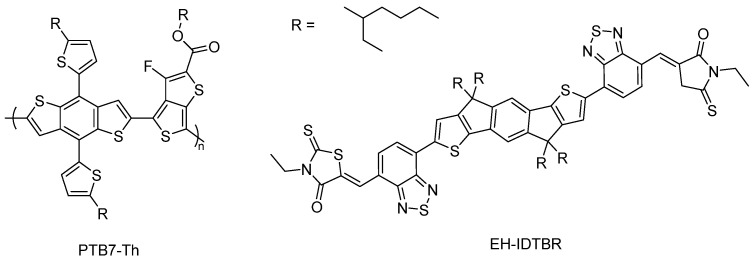
The structure of the donor (PTB7-Th) and the acceptor (EH-IDTBR) components used for the nanoparticle blends synthesized and tested for photocatalytic hydrogen production [30].

**Figure 39 nanomaterials-12-04299-f039:**
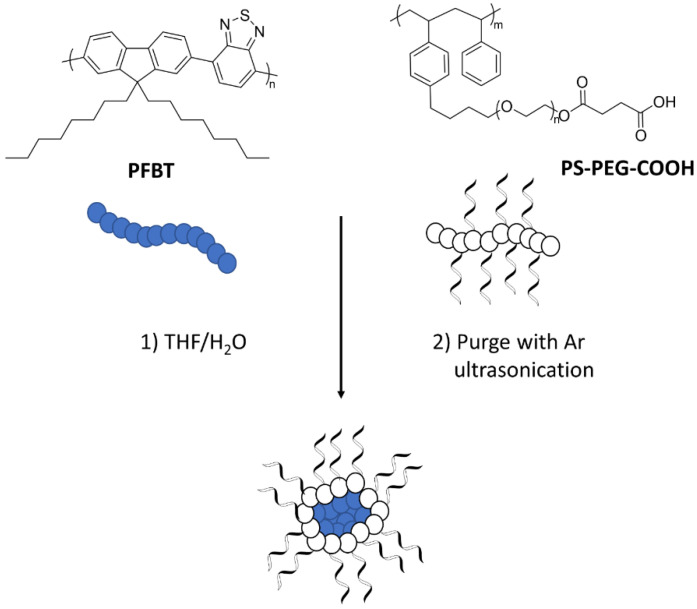
Synthetic outline of the PDots tested for photocatalytic hydrogen evolution [36].

**Figure 40 nanomaterials-12-04299-f040:**
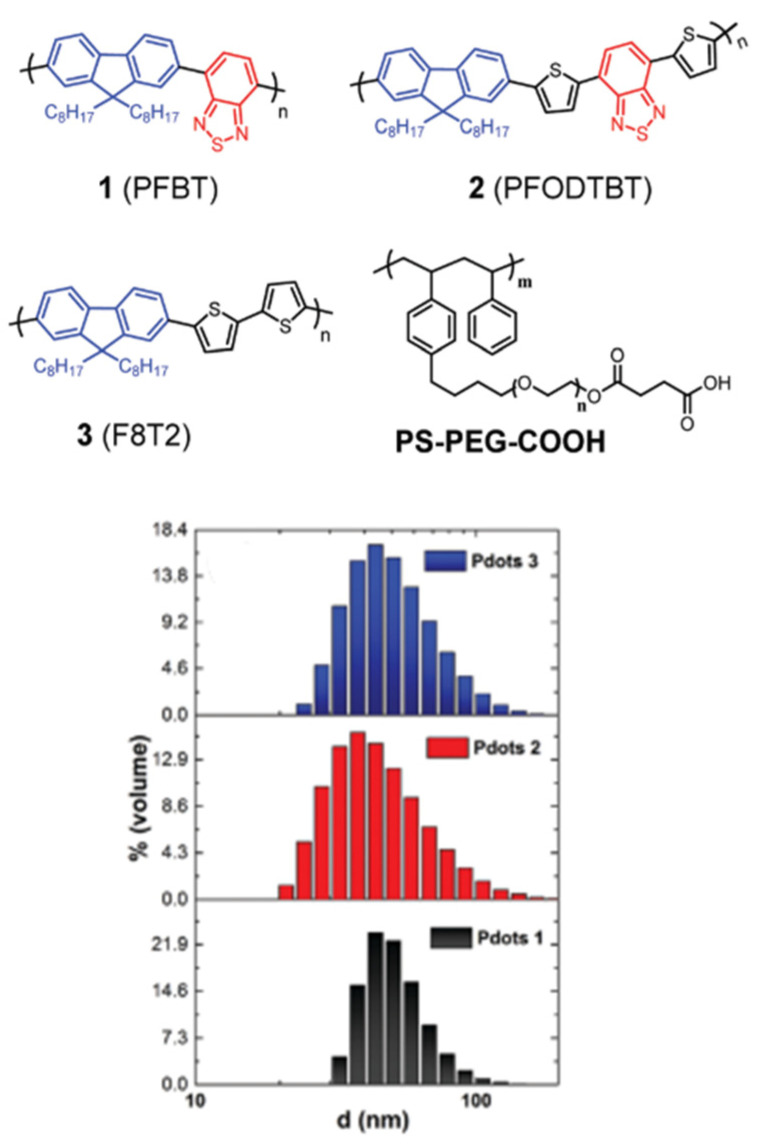
Donor-acceptor PDot structures with copolymer counterpart and dynamic light scattering (DLS) data indicating the relative size of the nanoparticles in each case. [Figure from ref. [159] under CC-BY 3.0 International License].

**Figure 41 nanomaterials-12-04299-f041:**
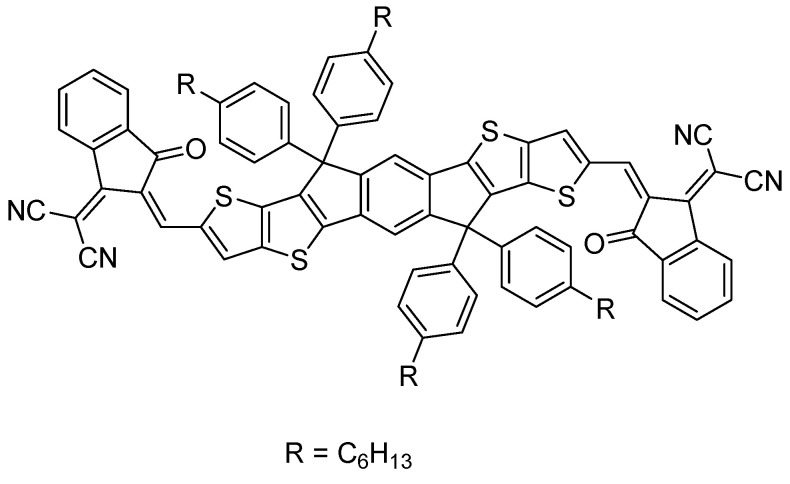
Structure of ITIC, the acceptor molecule utilized for the ternary PDot system [160].

**Figure 42 nanomaterials-12-04299-f042:**
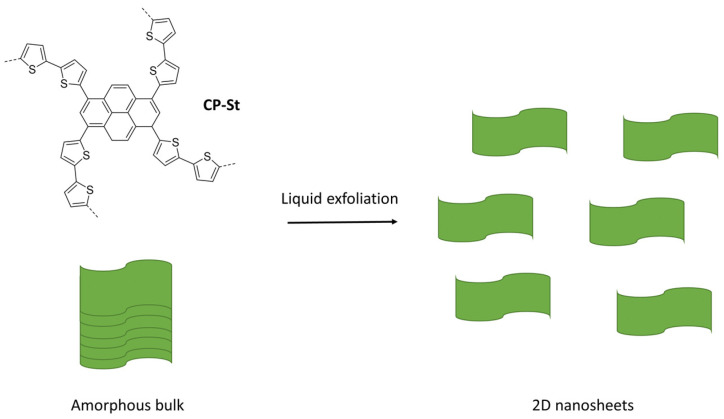
Structure of CP-St and a general scheme of the liquid exfoliation technique to produce 2D nanosheets [110].

**Figure 43 nanomaterials-12-04299-f043:**
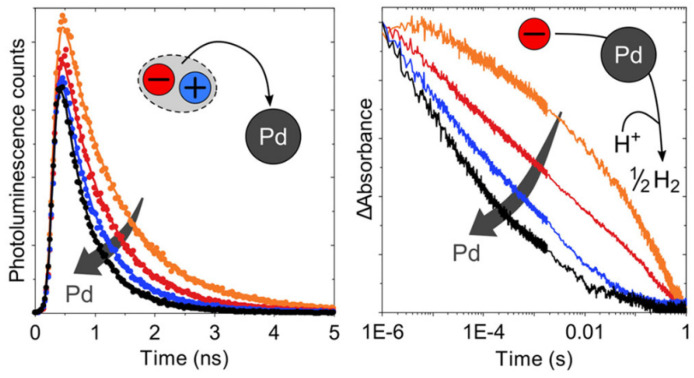
Time-resolved single photon counting data for F8BT (**left**) and TA data for P10 (**right**) for samples with increased palladium content [Figure from ref. [18] under CC-BY 3.0 International License].

**Figure 44 nanomaterials-12-04299-f044:**
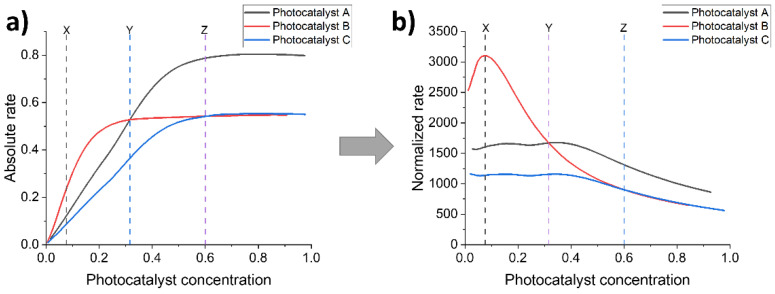
(**a**) Absolute and (**b**) mass normalized photocatalytic rates of catalysts.

**Table 1 nanomaterials-12-04299-t001:** Comparison of photocatalytic activity associated involving materials with polar cores and/or side-chains.

Organic Photocatalyst	Experimental Conditions	Light Source	HER (μmol h^−1^ g^−1^)	EQE (%)	Ref.
P10	25 mg photocatalyst; 25 mL H_2_O, MeOH, TEA (1:1:1); Pd 0.4 wt. %	300 W Xe, >420 nm filter	3260	11.6 (at 420 nm)	[48]
PyDTDO-3	10 mg photocatalyst; 90 mL 1.0 M ascorbic acid at pH 2.53, 10 mL DMF; Pd 16 ppm	300 W Xe, >420 nm filter	16,320	3.70 (at 420 nm)	[82]
PBDTTS-1SO	2 mg photocatalyst; 9 mL 1 M ascorbic acid at pH 4, 1 mL NMP; Pd 0.69 wt. % Pt 3 wt. % ^(b)^	350 W Xe, >380 nm filter	97,120	13.5 (at 420 nm)	[85]
PDPP3B-O4	25 mg photocatalyst; 60 mL H_2_O, TEA (5:1); Pd 0.20 wt. % Pt 1 wt. % ^(b)^	300 W Xe, >400 nm filter	5530	5.76 (at 450 nm)	[96]
FS-TEG	25 mg photocatalyst; 22.5 mL H_2_O, MeOH, TEA (1:1:1); Pd 0.3 wt. %	300 W Xe, >420 nm filter	2900	10 (at 420 nm)	[95]
PBDTBT-7EO	2.5 mg photocatalyst; 50 mL 0.2 M ascorbic acid at pH 4; Pt 3 wt. % ^(b)^ Residual Pd ^(a)^	300 W Xe, >420 nm filter	12,800	0.3 (at 600 nm)	[94]
PF6A-DBTO2	5 mg photocatalyst; 50 mL H_2_O, MeOH, TEA (3:1:1); Pd 1.37 wt. %	300 W Xe, >420 nm filter	21,930	8.28 (at 405 nm)	[99]
P-HEG-10	5 mg photocatalyst; 10 mL H_2_O, MeOH, TEA (1:1:1); Pt 5 wt. % ^(c)^ Pd 0.024 wt. %	300 W Xe, >380 nm filter	10,800	18.19 (at 420 nm)	[100]
PFBr-PhCN	2.5 mg photocatalyst; 50 mL 0.2 M ascorbic acid at pH 4; Pt 3 wt. % ^(b)^ Pd 1.32 wt. %	300 W Xe, filter not specified	15,320	0.42 (at 450 nm)	[111]
PSO-FNBr	2.5 mg photocatalyst 100 mL H_2_O, 7.5 vol% TEOA Pt 3 wt. % ^(b)^ Pd 0.21 wt. %	Xe ^(d)^, >420 nm filter	14,500	0.9 (at 450 nm)	[116]

^(a)^ Pd amount not specified. ^(b)^ Pt wt% value inferred in photodeposition. ^(c)^ Pt deposition process not specified. ^(d)^ Power not specified.

**Table 2 nanomaterials-12-04299-t002:** Comparison of photocatalytic activity associated involving materials with relatively large internal surface area.

Organic Photocatalyst	Experimental Conditions	Light Source	HER (μmol h^−1^ g^−1^)	EQE (%)	Ref.
SNP-BTT1	15 mg photocatalyst; 34 mL H_2_O, MeCN (1:1) at pH 7, 4 mL TEOA; Pd 0.61 wt. % Pt 3 wt. %^ (a)^	300 W Xe, >395 nm filter	3158	4.5 (at 420 nm)	[125]
Tr-F_3_N	10 mg photocatalyst; 50 mL H_2_O, ethylene glycol, TEOA (6:3:1); Pd 6.9 ppm%	300 W Xe, >300 nm filter	538	N.A.	[126]
Cp-St	6 mg photocatalyst; 30 mL 1 M ascorbic acid at pH 4, 6 mL NMP; Pd 0.739 wt. %, Sn 0.293 wt. %	300 W Xe, >420 nm filter	190,700	6.9 (at 550 nm)	[55]
S-CMP-3	25 mg photocatalyst; 25 mL H_2_O, MeOH, TEA (1:1:1); Pd 0.72 wt. %	300 W Xe, >420 nm filter	3106	13.2 (at 420 nm)	[37]
PTEPB	20 mg photocatalyst; 50 mL H_2_O; Cu 0.02 wt. %	300 W Xe, >420 nm filter	218	10.3 (at 420 nm)	[13]
TFPT-COF	10 mg photocatalyst; 10 mL H_2_O, TEOA (9:1); Pt 2.2 wt. % ^(a)^	300 W Xe, >420 nm filter	1970	3.9 (at 500 nm)	[134]
FS-COF	5 mg photocatalyst; 25 mL 0.1 M ascorbic acid at pH 2.6; 5 μL of 8 wt. % H_2_PtCl_6_ added ^(a)^	300 W Xe, >420 nm filter	10,100	3.2 (at 420 nm)	[143]

^(a)^ Pt wt. % value inferred in photodeposition. N.A.: Not Available.

**Table 3 nanomaterials-12-04299-t003:** Comparison of photocatalytic activity associated involving materials with relatively large internal surface area.

Organic Photocatalyst	Experimental Conditions	Light Source	HER (μmol h^−1^ g^−1^)	EQE (%)	Ref.
P10-e	325 μg photocatalyst; 25 mL aqueous [containing H_2_O and toluene (9:1), 10 mg mL^−1^ SDS and 3.5 mg mL^−1^ Na_2_CO_3_] MeOH, TEA (1:1:1) Pd 0.403 wt. %	300 W Xe, >420 nm filter	60,600	5.8 (at 420 nm)	[156]
PTB7-Th/ EH-IDTBR	2 mg photocatalyst (30:70 PT7B-Th:EH-IDTBR and TEBS surfactant); 20 mL 0.2 M ascorbic acid (pH 2); Pd 3129 ppm Pt 10% ^(a)^	300 W Xe, >350 nm filter	64,426	6.2 (at 700 nm)	[30]
PFODTBT/ PS-PEG-COOH	13 μg mL^−1^ photocatalyst; 3 mL 0.2 M ascorbic acid (pH 4); Pd 0.1 *w*/*w*%	White LED light (17 W, 5000 K), >420 nm filter	63,000	0.6 (at 550 nm)	[159]
D1,D2,ITIC/ PS-PEG- COOH	62 μg photocatalyst (55 wt. % ITIC); 2 mL 0.8 M ascorbic acid (pH 4); Pt 6 wt. % ^(b)^	White LED light (17 W, 5000 K), >420 nm filter	60,800	7.1 (at 600 nm)	[160]

^(a)^ Pt deposition process not specified. ^(b)^ Pt wt. % value inferred in photodeposition.

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
