# Peer review of "Impact of Interfaces, and Nanostructure on the Performance of Conjugated Polymer Photocatalysts for Hydrogen Production from Water"

_nanomaterials, 2022, doi:10.3390/nano12234299_

Round 1

Reviewer 1 Report

Manuscript is well written and all the sections are described comprehensively. There should be a difference between book section and a review article. I ll suggest authors to make this article more compact and focused to the point. The very basic information should be avoided and it should be focus on Indepth discussion of current issues and future perspectives.

Author Response

We thank the reviewer for their comments. As suggested, we have shortened the introduction and expanded the discussion and outlook sections.

Reviewer 2 Report

Recently, the problem of developing catalysts for the direct converting the energy of light into the energy of chemical bonds by splitting water and converting carbon dioxide has been actively discussed. In these processes the main role is given to nanostructured materials. This manuscript analyzes the current level of research on organic photocatalysts, which corresponds to the subject of the «Nanomaterials» journal. Particular emphasis is placed on the modification of organic photocatalysts, aimed at increasing their hydrophilicity. This problem is proposed to be solved by including new polar groups. The manuscript also describes examples of the use of polymer hydrogels for photocatalytic applications. The manuscript also describes examples of the use of polymer hydrogels for photocatalytic applications, which is a new strategy in the development of photocatalyst for water splitting. In addition, it was noted by the authors that little attention is paid in the literature to the role of the interface between the active component and the polymeric material. Nevertheless, there are such publications and they should be analyzed in this review.

In general, the manuscript reflects all trends in the study of polymer photocatalysts. I read it with great interest. I thank the authors for the written publication, based on a thorough analysis of a large number of studies performed by researchers from different countries.

I recommend this paper to be accepted for the publication with minor revision.

Comments

1). The role of the interface between the active component and the polymeric material should be discussed.

2). It is necessary to describe the principles for choosing (sacrificial) electron sources when using polymeric photocatalysts for water splitting.

Best regards, Reviewer.

Reviewer 3 Report

In this review, the authors addressed that, most of the semiconductors that drive the photocatalytic processes have been inorganic semiconductors but since the first report of carbon nitride organic semiconductors have also been considered. In this report, the authors focus on the impact of interfaces and nanostructuring on fundamental processes and performance in these organic photocatalysts. The review is well organized and contains interesting findings. However, I recommended a major revision of the article from its present form before it can be published in nanomaterials. The main concerns are listed below.

1. The abstract should be specific and scientific information.

2. The basic idea of the concept is appreciable. But, the authors should explain the novelty, need, and importance of the present review in the introduction part.

3. The figure’s quality in the review is not up to the standard.

4. The authors should explain the need for an organic photocatalysis process rather than inorganic photocatalysis.

5. The authors should explain the difficulties in the organic photocatalysis process.

6. The authors should mention the different experimental procedures for the preparation of these materials.

7. The authors should explain the challenges and future developments and not include them in the conclusion section.

8. The present report needs a tabulated analysis for photocatalytic hydrogen production.

9. In the current state, there are more typographical errors and the language should be improved.  Therefore, the authors are advised to recheck the whole manuscript for improving the language and structure carefully. 

Reviewer 4 Report

The manuscript edited "Impact of Interfaces and Nanostructure on the Performance of 1 Conjugated Polymer Photocatalysts for Hydrogen Production from Water"  deals with an important topic. 

Generally, the text is well described and proposed in a reader-friendly version.  I have a few minor questions, mostly related to editorial aspects. 

First, the main idea and topic should be covered in the introduction. Additionally, I recommend also giving some comparison with other similar reviews.  

From my perspective, using bold is not a consequence since not all chemicals are bold. 

 Conclusion: 

Please provide the perspective of ecological and human-related toxicological aspects of proposed nanomaterials. 

Round 2

Reviewer 3 Report

The manuscript can be acceptable in its present form.